# Decoupling Regularization and Privacy in Differentially Private Ridge Regression and ERM

**Wanjie Wang** [1]  **Tathagata Banerjee** [1]

## Abstract

We study ridge regression and ridge-regularized empirical risk minimization (ERM) under $(\varepsilon, \delta)$-differential privacy via output perturbation. In classical private ERM, the ridge parameter simultaneously controls statistical regularization and the estimator's global sensitivity. Larger regularization reduces the DP noise scale but increases bias. So choosing the tuning parameter becomes a privacy–accuracy bottleneck. We propose a framework that makes these two roles explicit by decoupling regularization into (i) a statistical penalty $\alpha$, defining the target ridge/ERM solution, and (ii) a privacy-stabilization parameter $c$, used only to enforce a curvature floor and hence a tight sensitivity bound. We apply this framework to ridge regression, where $c$ is used to boost the minimum eigenvalue of the empirical Gram matrix. We derive an explicit bias-variance-DP variance risk decomposition and characterize optimal $(\alpha, c)$ in several regimes, yielding sharp tuning guidance and improved accuracy relative to single-parameter regularization. Finally, we extend the same decoupling principle to general ridge-regularized ERM. We support the theory with simulations and applications on real data.

## 1. Introduction

In modern data analysis, the increasing availability of sensitive individual-level data has raised significant privacy concerns. Differential privacy (DP) has emerged as a rigorous framework for protecting individual information (Dwork, 2006; Dwork et al., 2014). DP requires that the distribution of an algorithm's output changes only slightly when a single individual's record is modified, providing protection against

---

arbitrary side information. While DP is an attractive default for releasing trained models and statistical summaries, it typically comes at the cost of an accuracy loss, motivating new algorithms that minimize this loss (Kifer et al., 2012; Huang et al., 2021).

**Differential privacy and sensitivity.** A classical way to achieve DP is to inject random noise calibrated to the sensitivity of an estimator. Let $Z = \{z_1, \ldots, z_n\}$ be a dataset and let $Z' = \{z_1, \ldots, z_i', \ldots, z_n\}$ be a neighboring dataset differing from $Z$ in exactly one record. An estimator $\tilde{\theta}(Z)$ is said to be $(\varepsilon, \delta)$-DP if for any measurable set $S$ of outputs,

$$\Pr(\tilde{\theta}(Z) \in S) \leq e^\varepsilon \Pr(\tilde{\theta}(Z') \in S) + \delta. \quad (1)$$

Given a non-private estimator $\hat{\theta}(Z)$, its global sensitivity is

$$\Delta_{\hat{\theta}} := \max_{Z \sim Z'} \|\hat{\theta}(Z) - \hat{\theta}(Z')\|.$$

A standard $(\varepsilon, \delta)$-DP estimator under Gaussian mechanism releases

$$\tilde{\theta}(Z) := \hat{\theta}(Z) + \frac{\Delta_{\hat{\theta}}\sqrt{2\log(1.25/\delta)}}{\varepsilon}\zeta, \ \zeta \sim \mathcal{N}(0, I), \quad (2)$$

so that smaller $\varepsilon$ requires larger injected noise (Dwork et al., 2014; Balle & Wang, 2018). This mechanism highlights the central challenge in private estimation: designing procedures with small sensitivity while retaining high statistical accuracy (Huang et al., 2021; Milionis et al., 2022; Cai et al., 2021).

**Ridge-regularized ERM and the tuning parameter.** This tension becomes particularly pronounced for ridge-regularized empirical risk minimization (ERM). Given data $Z = \{z_1, \ldots, z_n\}$, ridge-regularized ERM estimates parameters as

$$\hat{\theta}_\alpha(Z) \in \arg\min_{\theta \in \Theta} \frac{1}{n}\sum_{i=1}^n f(\theta; z_i) + \frac{\alpha}{2}\|\theta\|_2^2, \quad (3)$$

where $\alpha \geq 0$ is the regularization parameter. Under DP, $\alpha$ plays two competing roles. From a privacy perspective, larger $\alpha$ increases curvature and stabilizes the optimizer, typically reducing sensitivity $\Delta_{\hat{\theta}_\alpha}$ and hence the noise required

---

[1]Department of Statistics and Data Science, National University of Singapore, Singapore. Correspondence to: Wanjie Wang <wanjie.wang@nus.edu.sg>.

*Proceedings of the 43rd International Conference on Machine Learning*, Seoul, South Korea. PMLR 306, 2026. Copyright 2026 by the author(s).

for DP. From a statistical perspective, larger $\alpha$ increases bias and can substantially worsen estimation accuracy. Thus, $\alpha$ is no longer only a statistical tuning knob: it simultaneously controls regularization bias and the privacy noise scale.

**Risk metric.** We study the problem of selecting the ridge regularization level $\alpha$ in differentially private ridge-regularized ERM under a design-weighted parameter error. Our accuracy metric is the $\Sigma$-weighted parameter mean squared error (MSE),

$$\mathbb{E}\|\tilde{\theta} - \theta\|_\Sigma = \mathbb{E}[(\tilde{\theta} - \theta)^\intercal \Sigma (\tilde{\theta} - \theta)], \tag{4}$$

where $\Sigma$ captures the second-moment structure of predictors (e.g., $\Sigma = \mathbb{E}[xx^\intercal]$). Given privacy parameters $(\varepsilon, \delta)$ and problem dimensions $(n, p)$, we are interested in an $(\varepsilon, \delta)$-DP estimator with near-optimal MSE. Consider a special case that $\Sigma = I$, then the MSE caused by the extra DP noise is at the order of $O\left(\Delta_{\hat{\theta}_\alpha}^2 \cdot \frac{2\log(1.25/\delta)}{\varepsilon^2} \cdot p\right)$, where $\Delta_{\hat{\theta}_\alpha} \leq 1/\alpha$ assuming the loss function $f$ is convex and 1-Lipchitz. This observation justifies why a large $\alpha$ is preferred under DP. However, when $p \ll n$, a small $\alpha$ is preferred to reduce the variance of $\hat{\theta}_\alpha$, and the contradiction arises.

**Decoupling statistical regularization and sensitivity.** We address this challenge by introducing a DP framework that makes the two roles of regularization explicit. In our framework, the usual ridge tuning knob is decomposed into two parameters: the ridge regularization parameter $\alpha$, which governs the underlying ridge-regularized ERM solution; and a worst-case guard parameter $c$, which enforces an overall sensitivity bound and thereby reduces the DP noise scale. Here, $c$ is not a per-sample clipping threshold; rather, it provides a global guardrail that prevents pathological datasets (e.g., unusually ill-conditioned realizations) from inducing excessively large sensitivity and hence overwhelming DP noise. We give the decomposition of the MSE for the corresponding $(\varepsilon, \delta)$-DP estimator, where the DP noise contributes an accuracy loss with the denominator as $(c + \alpha)^2$ instead of $\alpha^2$. Hence, $\alpha$ can be chosen for statistical efficiency.

**Relation to LASSO and sparse regression.** Our work focuses on ridge regression and ridge-regularized ERM, where the closed-form structure of the optimizer allows explicit sensitivity analysis and regularization bias-DP variance decomposition. While LASSO and more general $\ell_1$-penalized estimators are a natural complement in high-dimensional sparse settings, they lack closed-form optimizers. Further, LASSO involves feature selection, which introduces additional privacy-related questions. Our decoupling idea can be extended in principle to LASSO and sparse regression settings, but the optimal parameter tuning in these setups requires separate treatment.

**Our contributions.** We instantiate this framework for ridge regression and analyze the expected $\Sigma$-weighted parameter MSE in three representative dimensional regimes: (i) $p \ll n$, (ii) $p = n^\beta$ for a constant $0 < \beta < 1$, and (iii) $p/n \to \gamma$ for a constant $\gamma > 0$. In these regimes we characterize the expected loss and show that, for appropriate choices of $(\alpha, c)$, the resulting rates match known minimax lower bounds in the corresponding settings (Cai et al., 2021). We additionally provide new results in the proportional asymptotics regime $p/n \to \gamma$. Beyond rate results, we give an exact characterization of the loss for general $(\alpha, c)$, which leads to explicit optimal choices of both parameters as a function of $n, p, \varepsilon, \delta$, and basic sample bounds. This removes the need for cross-validation-style tuning. Finally, we extend the same decoupling principle to more general ridge-regularized ERM objectives and support the theory with numerical evidence.

## 1.1. Related Work

In non-private penalized estimation, ridge regularization is classical and the tuning parameter $\alpha$ is typically selected to balance bias and variance (Zou & Hastie, 2005; ZHANG, 2010; Meanti et al., 2022). Standard practices include generalised cross-validation and efficient approximations to leave-one-out criteria (Patil et al., 2021; Tew et al., 2023). Complementing these procedures, sharp analyses characterize ridge regression and ridge-regularized ERM in high-dimensional regimes, including fundamental limits and proportional asymptotics (Taheri et al., 2021; Wu & Xu, 2020).

Differential privacy changes this picture because learning must be randomized in a way that is compatible with privacy constraints (Dwork, 2006; Dwork et al., 2014). A standard body of work studies private learning through ERM using broadly applicable mechanisms such as objective perturbation and output perturbation (Chaudhuri et al., 2011; Kifer et al., 2012). While these methods establish end-to-end privacy for broad classes of convex problems, they typically treat regularization as part of the learning rule rather than as an explicit object to be optimized for utility under a fixed privacy budget.

The tuning problem becomes especially delicate under DP. Searching over many candidate hyperparameters can consume privacy budget through repeated access to the data, and naive tuning can introduce privacy leakage if the selection procedure is not itself privatized. Prior work proposes stability-based validation approaches for private parameter selection (Chaudhuri & Vinterbo, 2013), and more recent work emphasizes practical risks and accounting considerations in private hyperparameter optimization (Koskela & Kulkarni, 2023). These results motivate approaches that either carefully privatize tuning or avoid extensive data-dependent search by deriving tuning rules with explicit de-

pendence on $(n, p, \varepsilon, \delta)$.

The contradiction between two roles of $\alpha$ in the worse case has been paid attention by (Wang, 2018). To reduce conservatism of worst-case calibration, (Wang, 2018) privately estimates the dataset sensitivity (e.g., $\lambda_{\min}(X^\mathsf{T} X)$)) and make proper adjustments. It does not provide a general two-parameter formulation separating statistical regularization from privacy stabilization, nor does it derive optimal tuning rules or sharp error characterizations across multiple $(n, p)$ regimes. The information-theoretic benchmarks have been established in (Cai et al., 2021) on the fundamental cost of privacy in high-dimensional regression, providing minimax lower bounds that our rates are compared against.

**Roadmap.** We start in Section 2 with the univariate case as an appetizer, which exposes the contradiction between statistically optimal ridge tuning and worst-case DP sensitivity, and motivates our decoupling idea via an additional stabilization parameter $c$. Section 3 develops the full multivariate decoupled adaptive DP ridge estimator with the decomposition of the $\Sigma$-weighted error. Section 4 discusses the parameter choice under proportional asymptotics. Section 5 extends the decoupling principle to general ridge-regularized ERM objectives and proves $(\varepsilon, \delta)$-DP. Empirical analysis on synthetic and real datasets can be found in Sections 6 and 7.

## 2. An Appetizer: Decoupled Adaptive DP Ridge Regression in Univariate Case

We begin with the simplest linear regression setting that $p = 1$ to illustrate the main tension of statistical optimization and DP sensitivity control, and demonstrate the role of our decoupling idea. In this case each $x_i$ is a scalar and

$$y_i = \theta x_i + \epsilon_i, \qquad i \in [n], \tag{5}$$

where $\theta$ is the unknown coefficient and $\epsilon_i$ is mean-zero noise. Our goal is to estimate $\theta$.

When $p = 1$, the ordinary least squares (OLS) estimator is

$$\hat{\theta}_{\mathrm{ols}}(x, y) = \frac{\sum_{i=1}^n x_i y_i}{\sum_{i=1}^n x_i^2},$$

and the ridge estimator with penalty parameter $\alpha \geq 0$ is

$$\hat{\theta}_\alpha(x, y) = \frac{\sum_{i=1}^n x_i y_i}{n\alpha + \sum_{i=1}^n x_i^2}.$$

When $n$ is large, OLS is unbiased and achieves the smallest variance among linear unbiased estimators, while ridge introduces bias. Thus, from a purely statistical perspective, the preferred choice in this univariate example is $\alpha \to 0$ when $n \to \infty$.

Now consider differential privacy. Let $Z = \{z_1, \ldots, z_n\}$ with $z_i = (x_i, y_i)$ and let $Z' = \{z_1, \ldots, z_i', \ldots, z_n\}$ be a neighboring dataset that differs in one record $z_i' = (x_i', y_i')$.

For concreteness, assume the data are bounded so that $|x_i| \leq 1$ and $|y_i| \leq 1$ almost surely (hence $|x_i y_i| \leq 1$ and $0 \leq x_i^2 \leq 1$). A direct calculation gives

$$\hat{\theta}_\alpha(Z) - \hat{\theta}_\alpha(Z') = \tag{6}$$

$$\frac{(x_i y_i - x_i' y_i') \sum_{j \neq i} x_j^2 + ((x_i')^2 - x_i^2) \sum_{j \neq i} x_j y_j}{(n\alpha + \sum_j x_j^2)(n\alpha + \sum_j (x_j')^2)}. \tag{7}$$

When $\alpha$ is a fixed constant, the denominator is of order $n^2$ while the numerator is of order $n$, leading to the familiar $O(1/n)$ sensitivity. Consequently, output perturbation yields an $(\varepsilon, \delta)$-DP release of the form

$$\tilde{\theta}_\alpha(Z) = \hat{\theta}_\alpha(Z) + \sigma_\alpha \zeta, \qquad \zeta \sim \mathcal{N}(0, 1), \tag{8}$$

with $\sigma_\alpha = \Omega(\frac{\sqrt{|\log \delta|}}{n\varepsilon})$ with $1/n$ as the global sensitivity.

The difficulty appears when the statistically preferred choice $\alpha \to 0$ is combined with the worst-case requirement of DP. Define the "good design" event

$$\mathcal{G}_{c_0} := \left\{ Z : \sum_{i=1}^n x_i^2 \geq c_0 n \right\}, \qquad c_0 \in (0, 1),$$

and the corresponding complement as $\mathcal{G}_{c_0}^c = \{Z : \sum_{i=1}^n x_i^2 < c_0 n\}$. On $\mathcal{G}_{c_0}$, even with $\alpha$ near zero, the denominator remains $\Omega(n^2)$ and the sensitivity remains $O(1/n)$, so the DP noise level is still at the optimal $O(1/(n\varepsilon))$ scale. However, DP requires the global sensitivity where $\max_{Z, Z'} \hat{\theta}_\alpha$. When $Z \in \mathcal{G}_{c_0}^c$ where $\sum_i x_i^2$ are very small, then the solution is highly unstable and increases the global sensitivity.

A simple example illustrates the issue. Consider $Z = \{x_1 = \tau, y_1 = 1, \ x_i \equiv 0, y_i \equiv 0 \ (i \geq 2)\}$, and $Z'$ differs from $Z$ only at $\{x_1' = -\tau, y_1' = 1\}$. Then $\sum_j x_j^2 = \tau^2$ and (6) gives

$$\Delta_{\hat\theta} \geq |\hat{\theta}_\alpha(Z) - \hat{\theta}_\alpha(Z')| \asymp \frac{1}{n\alpha + \tau^2}.$$

If $\alpha \to 0$ and $\tau^2$ is small, the global sensitivity $\Delta_{\hat\theta} \gg 1/n$, and the noise calibrated at $\Delta_{\hat\theta}/\varepsilon \gg 1/(n\varepsilon)$. This is the basic contradiction: statistically, we prefer small $\alpha$, whereas worst-case DP pushes toward larger $\alpha$ to control sensitivity.

**Decoupling.** The above contradiction is driven by rare but pathological datasets with extremely small $\sum_j x_j^2$ in the event $Z \notin \mathcal{G}_c$. Therefore, we introduce an extra parameter $c$ to protect the sensitivity from these cases. When $Z \notin \mathcal{G}_c$, we add $cn$ to the denominator to stabilize the estimator. Hence, $c$ actually takes the role of $\alpha$ to guard against the worst case.

For the univariate case, where the target ridge regularization $\alpha \geq 0$, define

$$\hat{\theta}_{\alpha,c}(Z) = \frac{\sum_{i=1}^{n} x_i y_i}{\alpha n + \max\{cn, \sum_{i=1}^{n} x_i^2\}}. \qquad (9)$$

We would like to choose $c$ so that $\mathcal{G}_c$ happens with high probability. For example, when $x_i$ follows subGaussian distributions with $\sigma^2 = 1$, $c = 1 - n^{-1/3}$ is a good choice. Therefore, $Z$ is well protected with a small noise and statistically optimal regularization $\alpha$, even when $\alpha \to 0$ and the worst case. This construction decouples the roles of $\alpha$ into $\alpha$ and $c$.

Under the boundedness assumption $|x_i| \leq 1$, $|y_i| \leq 1$, we have $\left|\sum_{i=1}^{n} x_i y_i\right| \leq n$ and $\alpha n + \max\{cn, \sum_{i=1}^{n} x_i^2\} \geq (\alpha + c)n$. Using a standard fraction-difference bound, for any neighboring $Z \sim Z'$,

$$\left|\hat{\theta}_{\alpha,c}(Z) - \hat{\theta}_{\alpha,c}(Z')\right| \leq \frac{4}{(\alpha + c)n} = \Delta_{\hat{\theta}_{\alpha,c}}.$$

Therefore, the Gaussian mechanism yields an $(\varepsilon, \delta)$-DP estimator, for $\zeta \sim \mathcal{N}(0, 1)$,

$$\tilde{\theta}_{\alpha,c}(Z) = \hat{\theta}_{\alpha,c}(Z) + \frac{4}{(\alpha + c)n} \cdot \frac{\sqrt{2 \log(1.25/\delta)}}{\varepsilon}. \qquad (10)$$

By the additional parameter $c$, the noise is controlled at the optimal rate at $O(1/(n\varepsilon))$ even when $\alpha \to 0$. Furthermore, $c$ does not affect the result when $Z \in \mathcal{G}_c$ with high probability, so it will not cause too heavy accuracy loss.

## 3. Decoupled Adaptive Differential Privacy Ridge Regression

### 3.1. Adaptive Differentially Private Ridge Estimator

We now turn to the multivariate setting. Let $x_i \in \mathbb{R}^p$ and consider the linear model

$$y_i = (\theta^*)^\intercal x_i + \epsilon_i, \qquad i \in [n], \qquad (11)$$

where $\epsilon_i$ are i.i.d. with mean 0 and variance 1. We measure estimation accuracy by the $\Sigma$-weighted parameter error

$$\|\theta - \theta^*\|_\Sigma^2 := (\theta - \theta^*)^\intercal \Sigma (\theta - \theta^*), \qquad (12)$$

which coincides with the excess prediction error when $\Sigma = \mathbb{E}[xx^\intercal]$ since $\mathbb{E}\left[(x^\intercal(\theta - \theta^*))^2\right] = \|\theta - \theta^*\|_\Sigma^2$.

Throughout this subsection, we assume the design covariance $\Sigma \succ 0$ is known. The unknown-$\Sigma$ case is deferred to Section 3.2. Assume $x_i$ are i.i.d. with mean 0 and covariance $\Sigma$, and that the signal is bounded in the prediction norm:

$$\|\theta^*\|_\Sigma = \|\Sigma^{1/2}\theta^*\|. \qquad (13)$$

Define the whitened parameter $\beta := \Sigma^{1/2}\theta$ and note that $\|\theta - \theta^*\|_\Sigma^2 = \|\beta - \beta^*\|_2^2$ with $\beta^* = \Sigma^{1/2}\theta^*$. To obtain a uniform DP sensitivity bound, we work with bounded (clipped) data. Fix radii $R_x, R_y > 0$ and define the projection operator

$$\Pi_a(v) := \frac{v}{(\|v\|_2/a) \vee 1}, \qquad a > 0.$$

We whiten and clip the covariates and clip the responses:

$$\tilde{x}_i := \Pi_{R_x}\left(\Sigma^{-1/2}x_i\right), \qquad \tilde{y}_i := \Pi_{R_y}(y_i), \qquad i \in [n].$$

Let $\tilde{X} = [\tilde{x}_1, \ldots, \tilde{x}_n] \in \mathbb{R}^{p \times n}$ and $\tilde{y} = (\tilde{y}_1, \ldots, \tilde{y}_n)$.

Now we introduce $c$ to this multivariate case. Let $Q \succeq 0$ be a PSD matrix. We define the uniform eigenvalue-boosting operator

$$Q \vee cI := Q + \left(c - \lambda_{\min}(Q)\right)_+ I, \qquad c > 0, \qquad (14)$$

so that $Q \vee cI \succeq cI$ always. This is a matrix analogue of the scalar guardrail in the univariate appetizer.

The following lemma shows that the boosting map is Lipschitz up to a constant factor.

**Lemma 3.1.** *For any PSD matrices $A, B \succeq 0$ and any $c > 0$,*

$$\|A \vee cI - B \vee cI\|_{\mathrm{op}} \leq 2\|A - B\|_{\mathrm{op}}.$$

**Estimator.** Given ridge level $\alpha \geq 0$ and stabilization level $c > 0$, define the stabilized matrix

$$\Sigma_{n,c,\alpha} := \left(\tilde{X}\tilde{X}^\intercal\right) \vee (nc)I + n\alpha I, \qquad (15)$$

and the (non-private) adaptive ridge estimator

$$\hat{\beta}_{\alpha,c}(Z) := \Sigma_{n,c,\alpha}^{-1} \tilde{X}\tilde{y}. \qquad (16)$$

Here $\alpha$ retains its statistical role, while $c$ is a dataset-level guardrail that prevents pathological realizations of $\tilde{X}\tilde{X}^\intercal$ from inflating sensitivity.

Since $\|\beta^*\|_2 = \|\theta^*\|_\Sigma \leq R_\theta$, we also project the estimator:

$$\hat{\beta}_{\alpha,c}^{\mathrm{proj}}(Z) := \Pi_{R_\theta}\left(\hat{\beta}_{\alpha,c}(Z)\right). \qquad (17)$$

Let $Z$ and $Z'$ be neighboring datasets differing in one sample. The following lemma shows that the projected estimator is uniformly stable, with stability controlled by $c + \alpha$:

**Lemma 3.2.** *The estimator $\hat{\beta}_{\alpha,c}^{proj}(Z)$ defined in (17) satisfies that, for all neighbouring datasets so that $\|x_i\|_2 \leq R_x$ and $|y_i| \leq R_y$, there is*

$$\left\|\hat{\beta}_{\alpha,c}^{\mathrm{proj}}(Z) - \hat{\beta}_{\alpha,c}^{\mathrm{proj}}(Z')\right\|_2 \leq \frac{R_x(4R_xR_\theta + 2R_y)}{n(c + \alpha)}. \qquad (18)$$

Therefore, the $(\varepsilon, \delta)$-DP estimator follows:

$$\tilde{\beta}_{\alpha,c}(Z) = \Pi_{R_\theta}\big(\hat{\beta}^{\mathrm{proj}}_{\alpha,c}(Z) + \sigma_{\alpha,c}\,\zeta\big), \quad \zeta \sim \mathcal{N}(0, I_p),$$

where

$$\sigma_{\alpha,c} = \frac{R_x(4R_x R_\theta + 2R_y)}{n(c+\alpha)} \cdot \frac{\sqrt{2\log(1.25/\delta)}}{\varepsilon}.$$

Mapping back to the original parameter gives

$$\tilde{\theta}_{\alpha,c}(Z) := \Sigma^{-1/2}\tilde{\beta}_{\alpha,c}(Z)$$

which remains $(\varepsilon, \delta)$-DP by post-processing.

Here are the results about our DADP estimator $\tilde{\theta}_{\alpha,c}$. It can be decomposed into the shrinkage and boosting bias, the sampling variance, and the variance from DP noise.

**Theorem 3.3.** *Assume $\Sigma$ is known and (13) holds. Let $\tilde{\theta}_{\alpha,c}$ be the output of Algorithm 1. Then $\tilde{\theta}_{\alpha,c}$ is $(\varepsilon, \delta)$-DP. Conditioning on the design matrix $\tilde{X}$ and $\theta^*$, there is*

$$\mathbb{E}\Big[\|\tilde{\theta}_{\alpha,c} - \theta^*\|^2_\Sigma \,\Big|\, \tilde{X}, \theta^*\Big] = \underbrace{\Big\|\Sigma^{-1}_{n,c,\alpha}\Big(\Sigma_{n,c,\alpha} - \tilde{X}\tilde{X}^\intercal\Big)\beta^*\Big\|^2_2}_{\text{(bias from ridge + boosting)}}$$

$$+ \underbrace{\mathrm{tr}\Big(\Sigma^{-1}_{n,c,\alpha}\tilde{X}\tilde{X}^\intercal\Sigma^{-1}_{n,c,\alpha}\Big)}_{\text{(sampling variance)}}$$

$$+ \underbrace{p\,\sigma^2_{\alpha,c}}_{\text{(DP variance)}} + o(n^{-1}),$$

*where $\beta^* = \Sigma^{1/2}\theta^*$ and $\sigma_{\alpha,c}$ is the noise scale in Algorithm 1.*

When the design matrix $\widetilde{X}$ is given, we already know whether $Z \in \mathcal{G}_c$ or not. If $Z \in \mathcal{G}_c$, then $\Sigma_{n,c,\alpha} = \tilde{X}\tilde{X}^\intercal + n\alpha I$, which is the standard ridge regression. The MSE is at the order of $O(\alpha^2 + p/n)$ for $p < n$. The DP variance term contributes $O(1/n^2\varepsilon^2(c+\alpha)^2)$. It is obvious that $c$ plays only in the DP variance term as a guardrail. This scenario happens with high probability if $c$ is chosen properly.

When $\lambda_{\min}(\widetilde{X}\widetilde{X}^\intercal) < c$, then $\Sigma_{n,c,\alpha} = \widetilde{X}\widetilde{X}^\intercal + n(\alpha + c - \lambda_{\min}(\widetilde{X}\widetilde{X}^\intercal))I$. Due to the boost in the ridge penalty, we have a larger MSE for ridge regression at $O((c - \lambda_{\min} + \alpha)^2 + p/n)$. The DP variance term stays the same. Hence, the extra accuracy cost by $c$ is at a constant level with small probability, hence it has very weak impact on the overall accuracy lost.

### 3.2. DADP Estimator with Inaccurate Covariance

In practice, the population covariance $\Sigma_x$ is unknown. We consider the setting where the covariates follow $x_i \sim \mathcal{N}(0, \Sigma_x)$. Here the Gaussian distribution can also be generalized to the sub-Gaussian distributions with covariance

---

**Algorithm 1** Decoupled Adaptive DP Ridge: $\tilde{\theta}_{\alpha,c}$

**Require:** Data $\{(x_i, y_i)\}^n_{i=1}$, privacy $(\varepsilon, \delta)$, ridge $\alpha \geq 0$, stabilization $c > 0$, $\Sigma \succ 0$, radii $R_x, R_y, R_\theta > 0$.
1: Define $\Pi_a(v) := v/((\|v\|_2/a) \vee 1)$.
2: Set $\tilde{x}_i \leftarrow \Pi_{R_x}(\Sigma^{-1/2}x_i)$, $\tilde{y}_i \leftarrow \Pi_{R_y}(y_i)$ for all $i$.
3: Form $\tilde{X} = [\tilde{x}_1, \ldots, \tilde{x}_n]$ and $\tilde{y} = (\tilde{y}_1, \ldots, \tilde{y}_n)^\intercal$.
4: Form $\Sigma_{n,c,\alpha} := (\tilde{X}\tilde{X}^\intercal) \vee (nc)I + n\alpha I$.
5: Compute $\hat{\beta}^{\mathrm{proj}}_{\alpha,c} \leftarrow \Pi_{R_\theta}(\Sigma^{-1}_{n,c,\alpha}\tilde{X}\tilde{y})$.
6: Draw $\zeta \sim \mathcal{N}(0, I_p)$ and set

$$\sigma_{\alpha,c} \leftarrow \frac{R_x(4R_x R_\theta + 2R_y)}{n(c+\alpha)} \cdot \frac{\sqrt{2\log(1.25/\delta)}}{\varepsilon},$$

$$\tilde{\beta}_{\alpha,c} \leftarrow \Pi_{R_\theta}\big(\hat{\beta}^{\mathrm{proj}}_{\alpha,c} + \sigma_{\alpha,c}\zeta\big).$$

7: Return $\tilde{\theta}_{\alpha,c} \leftarrow \Sigma^{-1/2}\tilde{\beta}_{\alpha,c}$.

---

matrix $\Sigma_x$ and uniformly bounded sub-Gaussian norm. We consider Gaussian distribution in theoretical analysis for simplicity. The algorithm is supplied with an external covariance proxy $\Sigma$ (e.g., estimated from a public or auxiliary dataset). In this subsection we investigate how using an inaccurate $\Sigma$ affects the utility of Algorithm 1.

Recall that the role of $\Sigma$ in Algorithm 1 is only to whiten the design $X$. Hence, an inaccurate estimate $\Sigma$ will affect the distribution of $\tilde{X}$. When the supplied covariance $\Sigma$ underestimates the true covariance $\Sigma_x$, i.e., $\Sigma_x \succeq \Sigma$, then the whitened covariates $\Sigma^{-1/2}x_i$ have covariance $\Sigma^{-1/2}\Sigma_x\Sigma^{-1/2} \succeq I$. Then the projection step follows to get $\tilde{x}_i$. Therefore, the global sensitivity bound does not change, and adding the same Gaussian noise guarantees $(\varepsilon, \delta)$-DP.

The only concern is the accuracy loss. According to the remark after Theorem 3.3, the accuracy loss is decided by the MSE of ridge regression and the DP variance. For inaccurate $\Sigma$, what will be affected is only the probability of two scenarios. In other words, we hope $\lambda_{\min}(\tilde{X}\tilde{X}^\intercal) < c$ happens with a small probability with the inaccurate $\Sigma$.

When $\Sigma \preceq \Sigma_x$, the covariance matrix of $\tilde{X}$ becomes larger than identity matrix in the PSD order, which makes small-eigenvalue pathologies less likely. Intuitively, if $\mathcal{G}_c$ happens under the correct whitening $(\Sigma_x)^{-1/2}X$, it will happen under $\Sigma^{-1/2}X$ with an underestimated $\Sigma$. Consequently $\mathcal{G}^c_c$, i.e., $\lambda_{\min}(\tilde{X}\tilde{X}^\intercal) < c$, occurs with smaller probability and the utility degradation due to stabilization is limited.

A final remark is on the estimation of $\Sigma$. Our algorithm and theoretical analysis are based on the whitened predictors $\Sigma^{-1/2}X$. The proofs, employing random matrix theory, requires the entries of $\Sigma^{-1/2}X$ to be independent. Hence, we assume an independent $\Sigma$ for simplicity. In practice, when

the covariance proxy $\Sigma$ is not available, one may consider using $X$ to directly estimate $\Sigma$ via cross-validation. In this case, we should consider subsamples or noise-perturbed covariance estimate $\hat{\Sigma}$ to protect the privacy. We leave it to further discussions.

We formalize this robustness in the following theorems. To start with, we consider the setting in (Cai et al., 2021) where the lower bound has been studied. In this setting, $\Sigma_x$ is scaled so that $p\Sigma_x$ is bounded.

**Theorem 3.4.** *Suppose there exists a constant $L$, so that $1/L \le p\lambda_{\min}(\Sigma_x) \le p\lambda_{\max}(\Sigma_x) \le L$. Suppose the true covariates satisfy $x_i \sim \mathcal{N}(0, \Sigma_x)$ and the algorithm is given a proxy covariance $\Sigma$ such that $\Sigma \preceq \Sigma_x \preceq C\Sigma$ for a constant $C \ge 1$.*

*Fix $R_x > 0$ as a constant, $R_\theta = \sqrt{p}$, and define $R_y = R_x\sqrt{p}$. Let $\alpha = 0$ and $c = 1 - (p/n)^{1/3}$, then our DADP estimator $\tilde{\theta}_{0,c}$ by Algorithm 1 with $\Sigma$ satisfies that*

- *$\tilde{\theta}_{0,c}$ is $(\varepsilon, \delta)$-DP.*

- *Moreover, when $p \ll n$, the true prediction loss is*

$$\mathbb{E}_{\Sigma_x}\left[\|\tilde{\theta}_{0,c} - \theta^*\|_{\Sigma_x}^2\right] \le C_1\frac{p}{n} + C_2\frac{R_x^4 p^2}{\varepsilon^2 n^2}\log\left(\frac{1.25}{\delta}\right),$$

*where $C_1, C_2$ depend only on $C$ and $L$, and are independent of $p, n, \varepsilon, \delta$.*

This rate matches the minimax lower bound in (Cai et al., 2021) up to constants. It shows the power of our algorithm.

We then consider a more general setting where $\Sigma_x$ does not shrink. In the following theorem, we consider both the low-dimensional case where $n \gg p$ and the high-dimensional case where $p/n \to \gamma \in (0, \infty)$ on some given $(\alpha, c)$ choices. We want to point out that, the minimax lower bound in (Cai et al., 2021) cannot be simply generalised to this setting by multiplying an extra $p$ in the loss function. The proof has to be revised substantially.

**Theorem 3.5.** *Assume the predictors satisfy $x_i \sim \mathcal{N}(0, \Sigma_x)$ and the algorithm is provided a proxy covariance $\Sigma$ such that $\Sigma \preceq \Sigma_x \preceq C\Sigma$ for a constant $C \ge 1$. Assume (13) holds with $R_\theta \le 1$. Fix $R_x = R_y = (p + \sqrt{2p\log n})^{1/2}$ for Algorithm 1 and denote the output as $\tilde{\theta}_{\alpha,c}$.*

*Then $\tilde{\theta}_{\alpha,c}$ is $(\varepsilon, \delta)$-DP. Moreover:*

- *Consider the case $n \gg p$. Choose $\alpha = 0$ and $c = 1 - (p/n)^{1/3}$. Then*

$$\mathbb{E}_{\Sigma_x}\left[\|\tilde{\theta}_{0,c} - \theta^*\|_{\Sigma_x}^2\right]$$
$$= O\left(\frac{p}{n} + \frac{p(p + \sqrt{2p\log n})^2}{n^2\varepsilon^2}\log(1.25/\delta)\right).$$

- *Suppose $p = n^\beta$ for $0 < \beta < 1$. Choose $c = 1 - (p/n)^{1/3}$, and $\alpha = \sqrt{p/n}$. Then*

$$\mathbb{E}_{\Sigma_x}\left[\|\tilde{\theta}_{\alpha,c} - \theta^*\|_{\Sigma_x}^2\right] = O\left(\frac{p}{n} + \frac{p^3\log(1.25/\delta)}{n^2\varepsilon^2}\right).$$

- *Suppose $p/n \to \gamma \in (0, \infty)$. For any $\alpha = \Omega(1)$ and $c = 0$,*

$$\mathbb{E}_{\Sigma_x}\left[\|\tilde{\theta}_{\alpha,0} - \theta^*\|_{\Sigma_x}^2\right] = O\left(1 + \frac{p^3\log(1.25/\delta)}{n^2\alpha^2\varepsilon^2}\right).$$

In the low-dimensional regime, we would prefer a regularization level $\alpha = 0$. With $c$ being at a constant level, the prediction loss caused by DP is at the order of $O(\frac{pR_x^2 R_y^2}{n^2\varepsilon^2})$, which matches the rate in (Wang, 2018; Sheffet, 2017). We want to remind readers that the rate is obtained when any design $X$ has a common conditional number, which is not required in our setting. When $p$ increases slowly, we increase $\alpha = \sqrt{p/n}$ to enjoy a better MLE for ridge regression. The accuracy lost is still at the same rate. When $p/n \to \gamma$, we would prefer a large $\alpha$ in this region as $\alpha = \Omega(1)$, then $\alpha$ itself already controls the stability, and $c$ can be taken as 0. Theorem 3.5 delivers a thorough theoretical result about $\tilde{\theta}_{\alpha,c}$ with respect to $n$ and $p$.

## 4. The Optimal Choices of $\alpha$ and $c$

Since the proposed estimator $\tilde{\theta}_{\alpha,c}$ is $(\varepsilon, \delta)$-DP for any $\alpha \ge 0$ and $c \ge 0$, it remains to choose $(\alpha, c)$ to minimize the prediction risk. To make this optimization explicit, we work under a well-specified random-design model and a Gaussian prior on the signal, following (Wu & Xu, 2020). Specifically, we assume

$$x_i \sim \text{sub}\mathcal{G}(0, \Sigma), \qquad \theta^* \sim \mathcal{N}\left(0, \frac{R_y^2}{p^2}I_p\right),$$

and consider the whitened setting where the empirical spectral distribution of $\frac{1}{n}\tilde{X}^\mathsf{T}\tilde{X}$ converges to the Marchenko–Pastur law with aspect ratio $\gamma = \lim_{n\to\infty} p/n \in (0, \infty)$:

$$\nu_\gamma(dx) = (1 - 1/\gamma)^+\delta_0 + \frac{\sqrt{(b_\gamma - x)(x - a_\gamma)}}{2\pi\gamma x}\mathbf{1}_{x\in[a_\gamma, b_\gamma]}\,dx,$$

where $a_\gamma = (1 - \sqrt{\gamma})^2$, $b_\gamma = (1 + \sqrt{\gamma})^2$. When $p \ll n$ (i.e., $\gamma \to 0$), the spectrum concentrates at 1, corresponding to the limit $\nu_\gamma \Rightarrow \delta_1$.

A key feature of our decoupling is that the effective ridge level that controls the statistical bias-variance trade-off is $\lambda(\alpha, c) := \alpha + (c - a_\gamma)_+$, while the privacy noise magnitude is controlled by the curvature lower bound $(\alpha + c)$ appearing in the sensitivity. Thus, $c$ can increase curvature (and reduce privacy noise) without necessarily increasing the effective ridge level $\lambda$, as long as $c \le a_\gamma$. This is formalized in Theorem 4.1.

**Theorem 4.1.** *Assume $p/n \to \gamma \in (0, \infty)$ and consider the proposed DPAR / DADP-Ridge estimator with parameters $(\alpha, c)$ in the (whitened) setting where the empirical eigenvalue distribution converges to the Marchenko–Pastur law $\nu_\gamma$ on $[a_\gamma, b_\gamma]$, with $a_\gamma = [(1-\sqrt{\gamma})_+]^2$ and $b_\gamma = (1+\sqrt{\gamma})^2$. Let $\lambda(\alpha, c) = \alpha + (c - a_\gamma)_+$, then the risk admits the decomposition*

$$g(\gamma, \alpha, c) := \mathbb{E}\big[\|\theta_{\alpha,c} - \theta^*\|_\Sigma^2\big]$$

$$\to \int_{a_\gamma}^{b_\gamma} \bigg( \underbrace{\frac{R_y^2}{p}\left(\frac{\lambda(\alpha, c)}{x + \lambda(\alpha, c)}\right)^2}_{(Bias)} + \underbrace{\frac{\gamma x}{(x + \lambda(\alpha, c))^2}}_{(Sampling\ Variance)} \bigg) \nu_\gamma(dx)$$

$$+ \underbrace{\frac{2\gamma^2 \left(R_x^2/p\right)\left(4R_x R_\theta + 2R_y\right)^2}{\varepsilon^2 (c + \alpha)^2} \log\left(\frac{1.25}{\delta}\right)}_{(DP\text{-}variance)}.$$

*Furthermore, if $\gamma < 1$, an optimal $(c, \alpha)$ will be obtained as $c = a_\gamma = (1 - \sqrt{\gamma})^2$.*

*Under the underparameterized setting, we have*

$$g(\gamma, \alpha, c) = \mathbb{E}[\|\theta_{\alpha,c} - \theta^*\|_\Sigma^2] \to$$

$$\underbrace{\frac{R_y^2}{p}\left(\frac{\alpha + (c-1)_+}{1 + \alpha + (c-1)_+}\right)^2}_{(Bias)} + \underbrace{\frac{p}{n(1 + \alpha + (c-1)_+)^2}}_{(\epsilon\text{-}Variance)}$$

$$+ \underbrace{\frac{2p\,R_x^2\left(4R_x R_\theta + 2R_y\right)^2}{n^2 \varepsilon^2 (c + \alpha)^2} \log\left(\frac{1.25}{\delta}\right)}_{(DP\text{-}variance)}.$$

*And the optimal parameters are given by*

$$c = 1, \quad \alpha = \frac{p^2}{n R_y^2} + \frac{2p\,R_x^2\left(4R_x R_\theta + 2R_y\right)^2}{\varepsilon^2 n^2} \log\left(\frac{1.25}{\delta}\right).$$

Figure 1 illustrates the phase behavior induced by $\lambda(\alpha, c) = \alpha + (c - a_\gamma)_+$. If $c < a_\gamma$, then $\lambda(\alpha, c) = \alpha$. Therefore, a small $\alpha$ will cause huge DP variance and hence the total risk. Increasing $\alpha$ causes a large decrease in DP-variance that dominates the whole term, so the curves are monotone decreasing in $\alpha$. If $c \in [a_\gamma, b_\gamma]$, then DP variance is properly controlled. Therefore, the statistical bias-variance trade-off related to $\alpha$ becomes active. The risk is typically non-monotone in $\alpha$, yielding an interior minimizer. When $c > b_\gamma$, the additional curvature behaves like excess regularization and inflates the bias. It elaborates how the decoupling works.

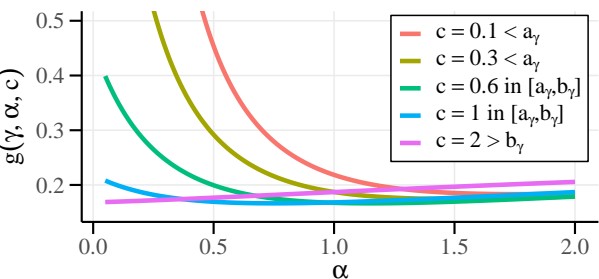

*Figure 1.* Risk function $g(\gamma, \alpha, c)$ versus the regularization level $\alpha$ for the stabilization parameter $c \in \{0.1, 0.3, 0.6, 1, 2\}$, given $\gamma = 0.1$. Different colors indicate different values of $c$. It can be found that $c = 1$ (light blue) achieves minimal risk.

**Choice of $\alpha$ and $c$ under generalisation.** To obtain explicit formulas for $\alpha$ and $c$ in Theorem 4.1, we need control on the lower tail of $\lambda_{\min}(\tilde{X}\tilde{X}^\top/n)$. For sub-Gaussian distributions, the choice of $\alpha$ and $c$ still holds. In our simulation and real data analysis, we apply this choice; see Sections 6–7. In more general settings beyond subgaussianity, the decoupling idea still applies, but a closed-form risk minimizer may not be available.

## 5. Generalisation to ERM

We provide a conceptual extension of the decoupling principle to general ERM problems. Here, we consider minimizing the empirical loss $L(\theta; Z) = \frac{1}{n}\sum_{i=1}^n f(\theta, z_i)$ for $\theta \in \Theta \subset \mathbb{R}^p$ over a bounded parameter domain $\Theta \subseteq \{\theta : \|\theta\| \leq R_\theta\}$. Regularization is often added to ensure strong convexity so that the minimizer is unique and stable, and can be computed by gradient-based methods.

We extend our decoupling idea to ERM by decoupling the regularization into the statistical tuning parameter $\alpha$ and an additional term $c$ that controls the solution's sensitivity to worst-case perturbations. Consider the ERM objective

$$L_{\alpha,c}(\theta; Z) = \frac{\sum_{i=1}^n f(\theta, z_i)}{n} + \frac{\|\theta\|^2}{2}\big(\alpha + (c - \frac{\sum_{i=1}^n h_f(z_i)}{n})_+\big),$$

where $h_f(z) = \min_{\theta \in \Theta} \lambda_{\min}(\nabla^2 f(\theta, z))$. Here $\lambda_{\min}$ matches the smallest eigenvalue of $\tilde{X}^\top \tilde{X}$ in the ridge regression case. We want to bound it so that the global sensitivity is well controlled. We demonstrate the pipeline in Algorithm 2.

Classical private ERM analyses typically assume $f(\cdot, z)$ is strongly convex for every $z$, which can be unrealistic when atypical observations produce nearly flat or even locally nonconvex losses. Our construction only requires that the average curvature of the sample (captured by $\frac{1}{n}\sum_i h_f(z_i)$) is not too negative. If it falls below $c$, the correction in $\alpha_c(Z)$

**Algorithm 2** Generalised DADP Estimator: $\tilde{\theta}_{\alpha,c}$

---

**Require:** Data $Z = \{z_i\}_{i=1}^n$, privacy $(\varepsilon, \delta)$, regularization $\alpha \geq 0$, stabilization $c > 0$, loss $f(\theta, z)$.

1: Compute $\hat{\theta}(Z) \leftarrow \arg\min_{\theta \in \Theta} L_{\alpha,c}(\theta; Z)$
2: Find the perturbation level of $L_{\alpha,c}(\theta; Z)$, which should be at $\sigma = O(1/(\alpha + c))$.
3: Draw $\zeta \sim \mathcal{N}(0, I_p)$ and set

$$\tilde{\theta}_{\alpha,c} \leftarrow \hat{\theta}(Z) + \frac{\sigma\sqrt{2\log(1.25/\delta)}}{\varepsilon}\zeta.$$

4: Return $\tilde{\theta}_{\alpha,c}$.

---

automatically adds curvature to ensure a uniform strong convexity floor $(\alpha + c)$ for $L_{\alpha,c}(\cdot; Z)$. In settings where the population loss is well-behaved, $\frac{1}{n}\sum_i h_f(z_i)$ concentrates around its population counterpart, so the correction is typically small and $\alpha$ continues to play the primary statistical role.

**Theorem 5.1.** *Algorithm 2 is $(\varepsilon, \delta)$-DP.*

## 6. Simulation studies

We conduct two simulation experiments to compare our proposed estimator with existing differentially private baselines. We examine (i) how the estimation error scales with the sample size $n$, and (ii) how the error changes as the privacy budget $\varepsilon$ increases. Throughout, we take $\Sigma = I_p$ and report the parameter estimation error

$$\text{Err}(\hat{\theta}) := \|\hat{\theta} - \theta_0\|_2^2,$$

averaged over 100 independent repetitions. The standard deviations are shown as error bars when the loss is not excessively large.

**Data generation.** We generate covariates $x_i \sim \mathcal{N}(0, I_p)$ independently and project them onto the Euclidean ball $x_i \leftarrow \Pi_{R_x}(x_i)$ with $R_x = \left(p + \sqrt{2p\log n}\right)^{1/2}$, which keeps almost all points unchanged in our regimes. We generate the ground-truth coefficients as $\theta^* \sim \mathcal{N}(0, \frac{1}{p}I_p)$, and then project it to $\theta^* \leftarrow \Pi_{R_\theta}(\theta^*)$ with $R_\theta = 1 + \sqrt{2\log(p)/p}$. Responses follow the well-specified linear model

$$y_i = \theta^\mathsf{T} x_i + \xi_i, \qquad \xi_i \sim \mathcal{N}(0, 1).$$

For methods requiring a response bound, we use $R_y = R_x R_\theta + \sqrt{\log n}$ which upper-bounds $\theta^\mathsf{T} x_i$ by $R_x R_\theta$ due to projection and controls the Gaussian noise term by a high-probability envelope.

We include the following baselines, all implemented with the recommended parameter settings in the original papers and $R_x, R_y, R_\theta$ in our setting.

- **CWZ** (Cai et al., 2021): optimal DP linear regression algorithms. We report the low-dimensional version (CWZ-LD) and the high-dimensional/sparse version (CWZ-HD).

- **Objective perturbation** (Kifer et al., 2012): DP objective perturbation ERM, with low-dimensional (Objpert-LD) and high-dimensional (Objpert-HD) implementations.

- **JL** (Sheffet, 2017): DP Johnson–Lindenstrauss projection estimator for least squares.

- **Our new method**. According to the theoretical results, we take

$$c = (1 - \sqrt{\gamma})^2 - \sqrt{\log n/n}, \qquad (19)$$

where $\gamma = p/n$. It is a finite-sample conservative version of $a_\gamma = (1-\sqrt{\gamma})^2$ suggested by the asymptotic analysis (Vershynin, 2010). Hence, $c < \lambda_{\min}(\widetilde{X}^\mathsf{T}\widetilde{X})$ most times. It gives us an approximation of the bias, sampling variance, and DP variance. Therefore, we can derive the optimal selection of $\alpha$ is that

$$\alpha = \frac{1}{c^2 R_\theta^2}\left(\frac{p}{n} + \frac{2pR_x^2\left(4R_x R_\theta + 2R_y\right)^2\log(1.25/\delta)}{n^2\varepsilon^2}\right). \qquad (20)$$

### 6.1. Estimation error versus sample size $n$

We consider two scaling regimes: (a) $p = n^{0.4}/2$ and (b) $p = n^{0.6}/5$, with $n \in \{1000, 3000, 5000, \ldots, 25000\}$. We fix $\varepsilon = 0.5$ and $\delta = 10/n$. Figure 2 reports the mean error and standard deviation over 100 repetitions. For methods with extremely large error (e.g., $> 100$), we omit the corresponding error bars since the standard deviations become visually uninformative.

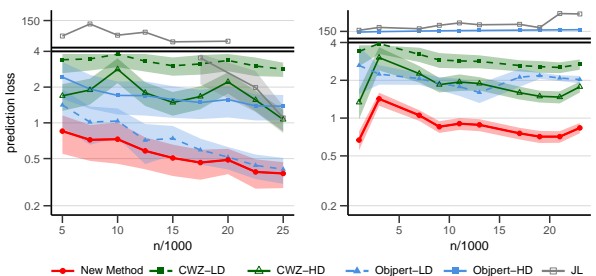

*Figure 2.* Parameter error $\|\hat{\theta} - \theta^*\|_2^2$ versus sample size $n$. Left: $p = n^{0.4}/2$. Right: $p = n^{0.6}/5$.

Figure 2 shows two consistent patterns. First, as $n$ increases, all the methods has an decreasing error, reflecting the improved statistical accuracy with more data. Second, our estimator achieves the smallest error across the full range

of $n$ in both regimes. The improvement is particularly pronounced in the more challenging regime $p = n^{0.6}/5$, where some baselines (notably Private-JL and CWZ-HD in our experiments) become unstable and incur very large error.

### 6.2. Estimation error versus privacy budget $\varepsilon$

We fix $n = 20000$ and consider two dimensions: $p = 70 \approx \sqrt{n}/2$ (left) and $p = 47 \approx \sqrt{n}/3$ (right). We fix $\delta = 10/n$ and vary $\varepsilon \in \{2^{-3}, 2^{-2}, 2^{-1}, 1, 2, 4, 8\}$. Figure 3 reports the mean error and standard deviation over 100 repetitions. As $\varepsilon$ increases (weaker privacy), all methods exhibit a decreasing error trend, consistent with the privacy-accuracy tradeoff. Standard deviations for cases with extremely large error are omitted.

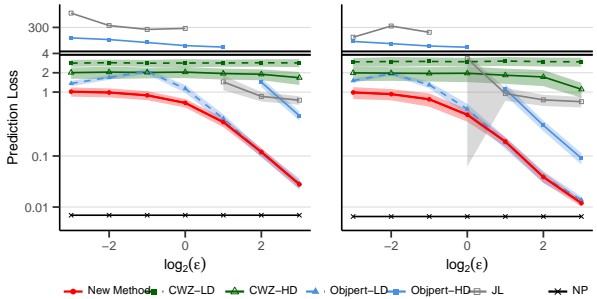

*Figure 3.* Parameter error $\|\hat{\theta} - \theta^*\|_2^2$ versus privacy budget $\varepsilon$. Left: $p = 70$. Right: $p = 47$.

Figure 3 shows that entering the less-private regime from the high-privacy regime, all methods benefit from a decreasing error as $\varepsilon$ increases. Across the entire privacy range, our method attains the smallest error, with the most visible gains in the high-privacy region (small $\varepsilon$), where privacy-induced noise dominates and the benefit of decoupling is most pronounced.

## 7. Real Data Analysis

We compare DP estimators on 4 datasets from the UCI Machine Learning Repository and the CMU StatLib repository: California house prices ($n = 14448$, $p = 8$) (Kelley Pace & Barry, 1997), Appliances Energy Prediction ($n = 13815$, $p = 27$) (Candanedo, 2017), Wine Quality ($n = 3429$, $p = 11$) (Cortez et al., 2009) and Boston Housing ($n = 355$, $p = 12$) (Harrison, D. and Rubinfeld, D.L., 1978). For each dataset, we take 30% of the dataset as auxiliary data and the other 70% for private regression. Details about datasets are in Appendix A.6.

We compare our DADP estimator with CWZ-HD, CWZ-LD, JL, ObjPert-HD and ObjPert-LD with $\varepsilon = 1$ and $\delta = 10/n$. We take the non-private estimate $\theta_0$ as the baseline. Define the error $Err(\hat{\theta}) = \|\hat{\theta} - \theta_0\|_\Sigma / \text{var}(y)$, where $\Sigma$ is calculated via the ancillary data. The ancillary dataset also decides

parameters for all algorithms, as described in Appendix A.6. The average error with standard deviation over 100 repetitions is reported in Table 1.

*Table 1.* Estimation errors of DP estimates on various data sets.

| Data | DADP | CWZ-HD | CWZ-LD | JL | ObjPert-HD | ObjPert-LD |
|------|------|--------|--------|-----|-----------|-----------|
| California | 0.31 (.09) | 36.4 (51) | 31.9 (34) | > 1e5 (3.1e6) | > 1e2 (14) | 1.42 (1.8) |
| energy | 0.11 (.01) | 1.75 (2.5) | 4.59 (4.1) | 0.51 (.68) | 0.30 (.05) | 0.14 (.11) |
| wine | 0.22 (.01) | 1.27 (1.4) | 10.8 (11) | > 1e3 (2.2e4) | 15.8 (13) | 1.05 (.61) |
| Boston | 0.63 (.02) | 26.2 (36) | 22.9 (22) | > 1e2 (1.6e3) | > 1e2 (130) | > 1e2 (280) |

Our estimator achieves the smallest error for all 4 datasets with the smallest standard deviation, supporting the broader applicability of the decoupling framework.

## Acknowledgements

This research was supported by the Singapore Ministry of Education Academic Research Fund Tier 1 A-8001451-00-00 and Tier 1 A-8003581-00-00. The authors also thank the anonymous reviewers for their valuable comments and suggestions.

## Impact Statement

This paper presents work whose goal is to advance the field of Differential Privacy in Machine Learning. We identify and resolve a core bottleneck in DP ridge-regularized ERM. The single regularization level $\alpha$ is forced to do two incompatible jobs: optimize statistical accuracy and suppress global sensitivity for privacy noise. We give a sharp bias–variance–DP-variance decomposition that exposes this conflict.

Building on this analysis, we introduce a principled two-parameter DP framework that decouples statistical regularization $\alpha$ from privacy stabilization $c$, yielding explicit, theory-driven tuning rules (no privacy-wasting hyperparameter sweeps) and better utility than classical one-parameter approaches, both theoretically and practically.

Our method can be used in domains where linear and ridge-type estimators are still the workhorse with privacy constraints, such as hospital and genomics studies, financial risk and fraud detection, telecom and platform analytics, and public-sector policy evaluation.

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

# A. Proof of main theorems

## A.1. Proof of Theorem 3.3

**Lemma A.1.** *For two vectors $v, v'$*

$$\left\| \frac{v}{\|v\| \vee a} - \frac{v'}{\|v'\| \vee a} \right\| \leq \left( \frac{\|v - v'\|}{(\|v'\| \wedge \|v\|) \vee a} \right)$$

*Proof.* Without loss of generality, we assume $\|v\| \leq \|v'\|$. Note that we can always divide all vectors by $a$, and normalize the problem of into showing

$$\left\| \frac{v}{\|v\| \vee 1} - \frac{v'}{\|v'\| \vee 1} \right\| \leq \left( \frac{\|v - v'\|}{\|v\| \vee 1} \right)$$

There are three scenarios. In the first scenario, $\|v\| \leq \|v'\| \leq 1$, then the claim is an identity. In the second scenario, $\|v\| \leq 1 \leq \|v'\|$. Then we want to show

$$\left\| v - \frac{v'}{\|v'\|} \right\| \leq \|v - v'\|,$$

in other words, the projection of point $v'$ unit ball is closer to $v$, a point in the unit ball, than the itself to the $v$. This is a well property due to the convexity of the unit ball.

Finally, if $1 < r = \|v\| \leq \|v'\| = br$ with $b \geq 1$. Suppose the angle between $v$ and $v'$ is $\theta$, then the left side is

$$\| \frac{v}{\|v\|} - \frac{v'}{\|v'\|} \| = 2 - 2\cos\theta = 4\sin(\frac{\theta}{2})^2.$$

The right hand side is

$$\frac{\|v - v'\|}{\|v\|} = 1 - 2\cos\theta b + b^2 =: g(b).$$

Since $\dot{g}(b) = 2b - 2\cos\theta \geq 0$ when $b \geq 1$, we find that $g(b) \geq g(0) =$ left hand side.

In conclusion, we have our claim.

$\square$

*Proof of Lemma 3.2.* Let $A = (\tilde{X}\tilde{X}^\top) \vee (nc)I + n\alpha I$ and $A' = (\tilde{X}'\tilde{X}'^\top) \vee (nc)I + n\alpha I$. Let $b = \tilde{X}\tilde{y}$ and $b' = \tilde{X}'\tilde{y}'$. Then

$$\hat{\beta}_{\alpha,c}(Z) - \hat{\beta}_{\alpha,c}(Z') = A^{-1}b - (A')^{-1}b' = (A')^{-1}(A' - A)\hat{\beta}_{\alpha,c}(Z) + (A')^{-1}(b - b').$$

Since $A \succeq n(c + \alpha)I$ and $A' \succeq n(c + \alpha)I$, we have $\|A^{-1}\|_{o_p} \leq 1/(n(c + \alpha))$ and $\|(A')^{-1}\|_{o_p} \leq 1/(n(c + \alpha))$. By Lemma 3.1,

$$\|A' - A\|_{o_p} \leq 2\|\tilde{X}'\tilde{X}'^\top - \tilde{X}\tilde{X}^\top\|_{o_p} = 2\|\tilde{x}_i'\tilde{x}_i'^\top - \tilde{x}_i\tilde{x}_i^\top\|_{o_p} \leq 4R_x^2,$$

and

$$\|b - b'\|_2 = \|\tilde{x}_i\tilde{y}_i - \tilde{x}_i'\tilde{y}_i'\|_2 \leq \|\tilde{x}_i\tilde{y}_i\|_2 + \|\tilde{x}_i'\tilde{y}_i'\|_2 \leq 2R_xR_y.$$

Thus

$$\|\hat{\beta}_{\alpha,c}(Z) - \hat{\beta}_{\alpha,c}(Z')\|_2 \leq \frac{4R_x^2\|\hat{\beta}_{\alpha,c}(Z)\|_2 + 2R_xR_y}{n(c + \alpha)}.$$

Applying Lemma A.1 with $a = R_\theta$ yields

$$\left\| \Pi_{R_\theta}(\hat{\beta}_{\alpha,c}(Z)) - \Pi_{R_\theta}(\hat{\beta}_{\alpha,c}(Z')) \right\|_2 \leq \frac{4R_x^2R_\theta + 2R_xR_y}{n(c + \alpha)} = \frac{R_x(4R_xR_\theta + 2R_y)}{n(c + \alpha)}.$$

This proves (18). $\square$

*Proof.* **(i) Privacy.** By Lemma 3.2, the projected estimator $\hat{\beta}^{\mathrm{proj}}_{\alpha,c}(Z) = \Pi_{R_\theta}(\hat{\beta}_{\alpha,c}(Z))$ has $\ell_2$-sensitivity

$$\Delta_2 \leq \frac{R_x(4R_xR_\theta + 2R_y)}{n(c + \alpha)}.$$

Algorithm 1 adds Gaussian noise $\sigma_{\alpha,c}\zeta$ with $\sigma_{\alpha,c} \geq \Delta_2\sqrt{2\log(1.25/\delta)}/\varepsilon$, hence $\hat{\beta}^{\mathrm{proj}}_{\alpha,c}(Z) + \sigma_{\alpha,c}\zeta$ is $(\varepsilon, \delta)$-DP by the Gaussian mechanism. The additional projection $\Pi_{R_\theta}$ and the back-transform $\Sigma^{-1/2}$ are post-processing, so the final $\tilde{\theta}_{\alpha,c}$ remains $(\varepsilon, \delta)$-DP.

**(ii) Decomposition.** Work in whitened coordinates, so that $\|\tilde{\theta}_{\alpha,c} - \theta^*\|^2_\Sigma = \|\tilde{\beta}_{\alpha,c} - \beta^*\|^2_2$. Ignoring the final projection (which is non-expansive and only decreases the squared distance), the released vector has the form

$$\tilde{\beta}_{\alpha,c} = \hat{\beta}_{\alpha,c} + \sigma_{\alpha,c}\zeta, \qquad \zeta \sim \mathcal{N}(0, I_p),$$

where

$$\hat{\beta}_{\alpha,c} = \Sigma^{-1}_{n,c,\alpha}\tilde{X}\tilde{y}.$$

On the regular event where clipping is inactive (which holds with probability $1 - O(n^{-1})$ under the stated model and our choice of $R_y$), we have $\tilde{y} = \tilde{X}^\top\beta^* + \epsilon$ with $\epsilon \sim \mathcal{N}(0, I_n)$ independent of $\zeta$. Hence

$$\hat{\beta}_{\alpha,c} - \beta^* = \Sigma^{-1}_{n,c,\alpha}(\tilde{X}\tilde{X}^\top - \Sigma_{n,c,\alpha})\beta^* + \Sigma^{-1}_{n,c,\alpha}\tilde{X}\,\epsilon.$$

Taking squared norms and conditional expectations given $(\tilde{X}, \theta^*)$, the cross term vanishes because $\mathbb{E}[\epsilon] = 0$, and

$$\mathbb{E}\big[\|\Sigma^{-1}_{n,c,\alpha}\tilde{X}\,\epsilon\|^2_2 \mid \tilde{X}\big] = \mathrm{tr}\Big(\Sigma^{-1}_{n,c,\alpha}\tilde{X}\tilde{X}^\top\Sigma^{-1}_{n,c,\alpha}\Big), \qquad \mathbb{E}\big[\|\sigma_{\alpha,c}\zeta\|^2_2\big] = p\,\sigma^2_{\alpha,c}.$$

The probability of clipping being active contributes an additional $o(n^{-1})$ term under our $R_x, R_y$ choices, yielding the stated decomposition. $\qquad\square$

### A.2. Proof of Theorem 3.4

*Proof.* Let $A := \Sigma^{-1/2}\Sigma_x\Sigma^{-1/2}$. The assumption $\Sigma \preceq \Sigma_x \preceq C\Sigma$ is equivalent to

$$I \preceq A \preceq CI.$$

We first relate the true prediction norm to the proxy norm: for any $v \in \mathbb{R}^p$,

$$\|v\|^2_{\Sigma_x} = v^\top\Sigma_x v = v^\top\Sigma^{1/2}A\Sigma^{1/2}v \leq \|A\|_{o_p}\,v^\top\Sigma v \leq C\,\|v\|^2_\Sigma.$$

Therefore,

$$\|\tilde{\theta}_{0,c} - \theta^*\|^2_{\Sigma_x} \leq C\,\|\tilde{\theta}_{0,c} - \theta^*\|^2_\Sigma. \tag{21}$$

It suffices to bound $\mathbb{E}\|\tilde{\theta}_{0,c} - \theta^*\|^2_\Sigma$.

Work in the whitened parameterization $\beta = \Sigma^{1/2}\theta$, so that $\|\theta - \theta^*\|^2_\Sigma = \|\beta - \beta^*\|^2_2$ and $\|\tilde{\theta}_{0,c} - \theta^*\|^2_\Sigma = \|\tilde{\beta}_{0,c} - \beta^*\|^2_2$. Let $u_i := \Sigma^{-1/2}x_i$. Then $u_i \sim \mathcal{N}(0, A)$ with $I \preceq A \preceq CI$.

Algorithm 1 clips the whitened covariates and responses so that $\|\tilde{x}_i\|_2 \leq R_x$ and $|\tilde{y}_i| \leq R_y$ deterministically, and then applies Gaussian output perturbation with sensitivity proportional to $R_x(4R_xR_\theta + 2R_y)/(n(c + \alpha))$. Thus the $(\varepsilon, \delta)$-DP guarantee holds by construction.

Condition on the realized (clipped) design $\tilde{X} = [\tilde{x}_1, \ldots, \tilde{x}_n]$. By Theorem 3.3 (risk decomposition), with $\alpha = 0$,

$$\mathbb{E}\Big[\|\tilde{\beta}_{0,c} - \beta^*\|^2_2 \,\Big|\, \tilde{X}, \beta^*\Big] = \underbrace{\Big\|\Sigma^{-1}_{n,c,0}\big(\Sigma_{n,c,0} - \tilde{X}\tilde{X}^\top\big)\beta^*\Big\|^2_2}_{\mathrm{Bias}(\tilde{X})} + \underbrace{\mathrm{tr}\Big(\Sigma^{-1}_{n,c,0}\tilde{X}\tilde{X}^\top\Sigma^{-1}_{n,c,0}\Big)}_{\mathrm{Var}(\tilde{X})} + p\,\sigma^2_{0,c}, \tag{22}$$

where $\Sigma_{n,c,0} = (\tilde{X}\tilde{X}^\top) \vee (nc)I$.

**DP term.** Algorithm 1 uses

$$\sigma_{0,c} = \frac{R_x\big(4R_x R_\theta + 2R_y\big)}{nc} \cdot \frac{\sqrt{2\log(1.25/\delta)}}{\varepsilon},$$

so

$$p\sigma_{0,c}^2 = \frac{2\,p\,R_x^2\big(4R_x R_\theta + 2R_y\big)^2}{n^2 c^2} \cdot \frac{\log(1.25/\delta)}{\varepsilon^2}.$$

With $R_\theta = \sqrt{p}$ and $R_y = R_x\sqrt{p}$, we have $(4R_x R_\theta + 2R_y)^2 = (5R_x\sqrt{p})^2 = 25R_x^2 p$. Since $c = 1 - (p/n)^{1/3} \to 1$ under $p \ll n$, it follows that

$$p\sigma_{0,c}^2 \le K_2'(C) \frac{R_x^4\,p^2}{n^2\varepsilon^2} \log\!\Big(\frac{1.25}{\delta}\Big), \tag{23}$$

for a constant $K_2'(C)$ depending only on $C$ and absolute constants.

**Bias and sampling variance.** Let $S := \tilde{X}\tilde{X}^\top/n$ and denote its eigenvalues by $\lambda_1 \ge \cdots \ge \lambda_p$. By construction of eigenvalue boosting,

$$\Sigma_{n,c,0} = n\big(S + (c - \lambda_p)_+ I\big) \succeq nS.$$

On the event $\{\lambda_p(S) \ge c\}$, boosting is inactive and hence $\Sigma_{n,c,0} = \tilde{X}\tilde{X}^\top$, so the bias term is exactly $0$. Moreover, in that case,

$$\mathrm{Var}(\tilde{X}) = tr\big((\tilde{X}\tilde{X}^\top)^{-1}\big) = \frac{1}{n}\sum_{i=1}^{p}\lambda_i^{-1} \le \frac{p}{n} \cdot \frac{1}{\lambda_p(S)} \le \frac{p}{nc} \le K_1'\frac{p}{n},$$

since $c \to 1$ under $p \ll n$.

It remains to control the probability that $\lambda_p(S) \ge c$. Because $\tilde{x}_i$ are bounded ($\|\tilde{x}_i\| \le R_x$) and satisfy $\mathbb{E}[u_i u_i^\top] = A$ with $I \preceq A \preceq CI$, standard concentration for bounded sub-Gaussian sample covariance implies that, with probability at least $1 - 2\exp(-c_0 p)$,

$$\|S - A\|_{o_p} \le C_0\sqrt{\frac{p}{n}},$$

for constants $C_0, c_0 > 0$ depending only on $R_x$ and $\|A\|_{o_p} \le C$. Hence on this event,

$$\lambda_p(S) \ge \lambda_{\min}(A) - \|S - A\|_{o_p} \ge 1 - C_0\sqrt{\frac{p}{n}}.$$

With the choice $c = 1 - (p/n)^{1/3}$ and $p \ll n$, we have $\sqrt{p/n} \ll (p/n)^{1/3}$ for $n$ large, so $\lambda_p(S) \ge c$ holds with probability at least $1 - 2e^{-c_0 p}$. The complement event contributes an exponentially small amount and can be absorbed into constants.

Taking expectation over $\tilde{X}$ in (22) and using the above bounds yields

$$\mathbb{E}\|\tilde{\beta}_{0,c} - \beta^*\|_2^2 \le K_1'(C)\frac{p}{n} + K_2'(C)\frac{R_x^4 p^2}{n^2\varepsilon^2} \log\!\Big(\frac{1.25}{\delta}\Big).$$

Since $\|\tilde{\theta}_{0,c} - \theta^*\|_\Sigma^2 = \|\tilde{\beta}_{0,c} - \beta^*\|_2^2$, combining with (21) gives

$$\mathbb{E}\|\tilde{\theta}_{0,c} - \theta^*\|_{\Sigma_x}^2 \le C\,\mathbb{E}\|\tilde{\theta}_{0,c} - \theta^*\|_\Sigma^2 \le K_1(C)\frac{p}{n} + K_2(C)\frac{R_x^4 p^2}{n^2\varepsilon^2} \log\!\Big(\frac{1.25}{\delta}\Big),$$

which completes the proof. $\qquad\square$

### A.3. Proof of Theorem 3.5

*Proof.* Since $\Sigma_x \preceq C\Sigma$, for any $v \in \mathbb{R}^p$ we have $v^\top \Sigma_x v \le C\,v^\top \Sigma v$, hence

$$\|\tilde{\theta}_{\alpha,c} - \theta^*\|_{\Sigma_x}^2 \le C\,\|\tilde{\theta}_{\alpha,c} - \theta^*\|_\Sigma^2. \tag{24}$$

It suffices to bound $\mathbb{E}\|\tilde{\theta}_{\alpha,c} - \theta^*\|_\Sigma^2$.

Let $\beta^* = \Sigma^{1/2}\theta^*$ and recall the algorithm outputs $\tilde{\theta}_{\alpha,c} = \Sigma^{-1/2}\tilde{\beta}_{\alpha,c}$, so

$$\|\tilde{\theta}_{\alpha,c} - \theta^*\|_\Sigma^2 = \|\tilde{\beta}_{\alpha,c} - \beta^*\|_2^2.$$

Moreover, $\Sigma \preceq \Sigma_x$ implies $\|\beta^*\|_2 = \|\Sigma^{1/2}\theta^*\| \leq \|\Sigma_x^{1/2}\theta^*\| \leq 1$.

Write $u_i := \Sigma^{-1/2}x_i$. Then $u_i \sim \mathcal{N}(0, A)$ with

$$A := \Sigma^{-1/2}\Sigma_x\Sigma^{-1/2}, \qquad I \preceq A \preceq CI.$$

Define the (no-clipping) event

$$\mathcal{E} := \left\{ \max_{i\in[n]} \|u_i\|_2 \leq R_x, \ \max_{i\in[n]} |y_i| \leq R_y \right\}.$$

On $\mathcal{E}$, the projections do not activate and thus $\tilde{x}_i = u_i$ and $\tilde{y}_i = y_i$ for all $i$.

Conditioning on $\tilde{X}$, Theorem 3.3 yields

$$\mathbb{E}\left[ \|\tilde{\beta}_{\alpha,c} - \beta^*\|_2^2 \,\Big|\, \tilde{X}, \beta^* \right] = \left\| \Sigma_{n,c,\alpha}^{-1}(\Sigma_{n,c,\alpha} - \tilde{X}\tilde{X}^\top)\beta^* \right\|_2^2 + tr\left( \Sigma_{n,c,\alpha}^{-1}\tilde{X}\tilde{X}^\top\Sigma_{n,c,\alpha}^{-1} \right) + p\,\sigma_{\alpha,c}^2. \tag{25}$$

Let $S := \tilde{X}\tilde{X}^\top/n$ with eigenvalues $\lambda_1 \geq \cdots \geq \lambda_p$. Since

$$(\tilde{X}\tilde{X}^\top) \vee (nc)I = \tilde{X}\tilde{X}^\top + n(c - \lambda_p)_+I,$$

we have

$$\Sigma_{n,c,\alpha} = n\Big( S + \alpha I + (c - \lambda_p)_+I \Big).$$

Hence $\Sigma_{n,c,\alpha} - \tilde{X}\tilde{X}^\top = n(\alpha + (c - \lambda_p)_+)I$, and with $\|\beta^*\|_2 \leq 1$,

$$\left\| \Sigma_{n,c,\alpha}^{-1}(\Sigma_{n,c,\alpha} - \tilde{X}\tilde{X}^\top)\beta^* \right\|_2^2 \leq \Big( 1 + \frac{\lambda_p}{\alpha + (c - \lambda_p)_+} \Big)^{-2} \leq 1. \tag{26}$$

Also, diagonalizing $S$ gives

$$tr\Big( \Sigma_{n,c,\alpha}^{-1}\tilde{X}\tilde{X}^\top\Sigma_{n,c,\alpha}^{-1} \Big) = \frac{1}{n}\sum_{i=1}^{p} \frac{\lambda_i}{(\lambda_i + \alpha + (c - \lambda_p)_+)^2}. \tag{27}$$

For the DP term, Algorithm 1 uses

$$\sigma_{\alpha,c} = \frac{R_x\big(4R_xR_\theta + 2R_y\big)}{n(c+\alpha)} \cdot \frac{\sqrt{2\log(1.25/\delta)}}{\varepsilon},$$

so

$$p\sigma_{\alpha,c}^2 = \frac{2\,p\,R_x^2\big(4R_xR_\theta + 2R_y\big)^2}{n^2(c+\alpha)^2} \cdot \frac{\log(1.25/\delta)}{\varepsilon^2}. \tag{28}$$

With $R_x = R_y = (p + \sqrt{2p\log n})^{1/2}$ and $R_\theta \leq 1$,

$$\big(4R_xR_\theta + 2R_y\big)^2 \leq (4R_x + R_x)^2 = 25R_x^2,$$

so the DP term satisfies

$$p\sigma_{\alpha,c}^2 = O\left( \frac{p\,R_x^4}{n^2(c+\alpha)^2} \cdot \frac{\log(1.25/\delta)}{\varepsilon^2} \right) = O\left( \frac{p\big(p + \sqrt{2p\log n}\big)^2}{n^2(c+\alpha)^2\varepsilon^2} \log(1.25/\delta) \right).$$

It remains to control $\Pr(\mathcal{E}^c)$ and then specialize to regimes. Since $u_i \sim \mathcal{N}(0, A)$ with $A \preceq CI$, a standard Gaussian quadratic-form tail bound implies that for any $t > 0$,

$$\Pr\Big( \|u_i\|_2^2 \geq tr(A) + 2\sqrt{tr(A^2)t} + 2\|A\|_{op}t \Big) \leq e^{-t}.$$

Taking $t = 2 \log n$ and using $tr(A) \asymp p$, $tr(A^2) \lesssim p$, $\|A\|_{o_p} \leq C$, we obtain

$$\Pr\Big( \|u_i\|_2^2 \geq c_0 \big( p + \sqrt{p \log n} + \log n \big) \Big) \leq n^{-2},$$

for a constant $c_0$ depending only on $C$. By enlarging radii by absolute constants and lower-order terms (as permitted), we may assume $R_x^2 \geq c_0(p + \sqrt{p \log n} + \log n)$, so that $\Pr(\max_i \|u_i\|_2 > R_x) \leq n^{-1}$. Similarly, since $y_i = \beta^{*\top} u_i + \epsilon_i$ with $\|\beta^*\|_2 \leq 1$, we have $|\beta^{*\top} u_i| \leq \|u_i\|_2$; combining the tail of $\|u_i\|_2$ with a Gaussian tail for $\epsilon_i$ and enlarging $R_y$ by constants if needed gives $\Pr(\max_i |y_i| > R_y) \leq n^{-1}$. Therefore $\Pr(\mathcal{E}^c) \leq 2n^{-1}$. On $\mathcal{E}^c$, the projection ensures $\|\hat{\beta}_{\alpha,c}^{\text{proj}}\|_2 \leq R_\theta \leq 1$, so $\mathbb{E}[\|\tilde{\beta}_{\alpha,c} - \beta^*\|_2^2 \mathbf{1}_{\mathcal{E}^c}]$ is $o(1)$ compared to the dominant rates and can be absorbed into the $O(\cdot)$ terms.

Low-dimensional regime $n \gg p$. Work on $\mathcal{E}$, so $\tilde{X} = [u_1, \dots, u_n]$ and $S = (1/n) \sum u_i u_i^\top$. A standard matrix concentration inequality gives

$$\|S - A\|_{o_p} \leq C_1 \Big( \sqrt{\frac{p}{n}} + \frac{p}{n} \Big)$$

with probability at least $1 - 2e^{-c_1 p}$, for constants $C_1, c_1 > 0$ depending only on $C$. Since $\lambda_{\min}(A) \geq 1$, we have $\lambda_p(S) \geq 1 - C_1(\sqrt{p/n} + p/n)$ on this event. Choosing $\alpha = 0$ and $c = 1 - (p/n)^{1/3}$ and taking $n \gg p$, we have $\lambda_p(S) \geq c$ and thus $(c - \lambda_p)_+ = 0$; hence the bias term vanishes. Moreover, with $\alpha = 0$ and boosting inactive,

$$\text{Var}(\tilde{X}) = \frac{1}{n} \sum_{i=1}^p \frac{1}{\lambda_i} = tr\big( (\tilde{X}\tilde{X}^\top)^{-1} \big).$$

Since $\tilde{X}\tilde{X}^\top \sim \text{Wishart}(A, n)$ and $n > p + 1$,

$$\mathbb{E}\big[ (\tilde{X}\tilde{X}^\top)^{-1} \mathbf{1}_{\mathcal{E}} \big] \leq \frac{A^{-1}}{n - p - 1}, \qquad \mathbb{E}\big[ \text{Var}(\tilde{X}) \mathbf{1}_{\mathcal{E}} \big] \leq \frac{tr(A^{-1})}{n - p - 1} \leq \frac{p}{n - p - 1} = O\Big( \frac{p}{n} \Big),$$

using $A \succeq I$.

On the event $\mathcal{E}^c$, $\tilde{\beta}_{\alpha,c}$ and $\beta^*$ are bounded, so the loss $\mathbb{E}[\|\tilde{\beta}_{\alpha,c} - \beta^*\|_2^2]$ is also bounded. Hence, $\mathbb{E}[\|\tilde{\beta}_{\alpha,c} - \beta^*\|_2^2 \mathbf{1}_{\mathcal{E}^c}] \lesssim \Pr(\mathcal{E}^c) \lesssim 1/n$. Combining the two parts and it follows that

$$\mathbb{E}\big[ (\tilde{X}\tilde{X}^\top)^{-1} \big] = \frac{A^{-1}}{n - p - 1}, \qquad \mathbb{E}\, \text{Var}(\tilde{X}) = \frac{tr(A^{-1})}{n - p - 1} \leq \frac{p}{n - p - 1} = O\Big( \frac{p}{n} \Big),$$

Combining this with (28) (with $c \asymp 1$) yields the first bullet.

Proportional regime $p/n \to \gamma \in (0, \infty)$. For $c = 0$ and $\alpha = \Omega(1)$, (26) gives $\text{Bias}(\tilde{X}) \leq 1$. From (27) and $\lambda/(\lambda + \alpha)^2 \leq \lambda/\alpha^2$,

$$\text{Var}(\tilde{X}) \leq \frac{1}{n\alpha^2} \sum_{i=1}^p \lambda_i = \frac{1}{n\alpha^2} tr(S) = \frac{1}{n\alpha^2} \cdot \frac{1}{n} \sum_{i=1}^n \|u_i\|_2^2.$$

Taking expectations gives $\mathbb{E}\, \text{Var}(\tilde{X}) = tr(A)/(n\alpha^2) \leq Cp/(n\alpha^2) = O(\gamma/\alpha^2) = O(1)$. Adding (28) (with $c = 0$) yields the third bullet.

For the refined choice when $\gamma < 1$, take $c = a_\gamma - 1/\sqrt{n}$ and $\alpha = \frac{a_\gamma}{\gamma^{-1} - 1}$. Standard smallest-eigenvalue results for $p/n \to \gamma < 1$ imply $\lambda_p(S) \geq a_\gamma - o_{\P}(1)$, so $\Pr(\lambda_p(S) \geq c) \to 1$ and boosting is inactive w.h.p., hence $\text{Bias}(\tilde{X}) \leq 1$ by (26). Moreover, on the event $\{\lambda_p(S) \geq c\}$,

$$\text{Var}(\tilde{X}) = \frac{1}{n} \sum_{i=1}^p \frac{\lambda_i}{(\lambda_i + \alpha)^2} \leq \frac{1}{n} \sum_{i=1}^p \frac{\lambda_i}{(\lambda_i + c + \alpha)^2} \leq \frac{1}{n} \sum_{i=1}^p \frac{\lambda_i}{(c + \alpha)^2} = \frac{1}{(c + \alpha)^2} \cdot \frac{1}{n} tr(S).$$

Taking expectations gives $\mathbb{E}\, \text{Var}(\tilde{X}) = O(\gamma/(c + \alpha)^2) = O(1)$ (since $c + \alpha$ is bounded away from 0 for fixed $\gamma < 1$). Adding (28) completes the refined bound.

Finally, combining with (24) proves the theorem. $\qquad \square$

## A.4. Proof of Theorem 4.1

*Proof.* We work in the whitened parameterization (so that $\|\theta - \theta^*\|_\Sigma^2$ equals the squared Euclidean error of the whitened parameter), and condition on the realized design matrix used by the estimator. By Theorem 3.3, conditioning on $\tilde{X}$, the risk decomposes into a bias term, an $\epsilon$-variance term, and a DP-variance term:

$$\mathbb{E}\Big[\|\theta_{\alpha,c} - \theta^*\|_\Sigma^2 \,\Big|\, \tilde{X}, \theta^*\Big] = \left\|\Sigma_{n,c,\alpha}^{-1}\big(\Sigma_{n,c,\alpha} - \tilde{X}\tilde{X}^\top\big)\beta^*\right\|_2^2 + tr\Big(\Sigma_{n,c,\alpha}^{-1}\tilde{X}\tilde{X}^\top\Sigma_{n,c,\alpha}^{-1}\Big) + p\sigma_{\alpha,c}^2 + o(1), \tag{29}$$

where $\beta^* = \Sigma^{1/2}\theta^*$ and $\Sigma_{n,c,\alpha} = (\tilde{X}\tilde{X}^\top) \vee (nc)I + n\alpha I$.

Let $S := \tilde{X}\tilde{X}^\top/n$ with eigenvalues $\lambda_1 \geq \cdots \geq \lambda_p$. By definition of the boosting operator,

$$(\tilde{X}\tilde{X}^\top) \vee (nc)I = \tilde{X}\tilde{X}^\top + n(c - \lambda_p)_+ I,$$

hence

$$\Sigma_{n,c,\alpha} = n\Big(S + \alpha I + (c - \lambda_p)_+ I\Big).$$

Therefore $\Sigma_{n,c,\alpha} - \tilde{X}\tilde{X}^\top = n(\alpha + (c - \lambda_p)_+)I$, and diagonalizing $S$ yields the exact forms

$$\left\|\Sigma_{n,c,\alpha}^{-1}\big(\Sigma_{n,c,\alpha} - \tilde{X}\tilde{X}^\top\big)\beta^*\right\|_2^2 = \sum_{i=1}^{p}\left(\frac{\alpha + (c - \lambda_p)_+}{\lambda_i + \alpha + (c - \lambda_p)_+}\right)^2 \langle\beta^*, v_i\rangle^2, \tag{30}$$

$$tr\Big(\Sigma_{n,c,\alpha}^{-1}\tilde{X}\tilde{X}^\top\Sigma_{n,c,\alpha}^{-1}\Big) = \frac{1}{n}\sum_{i=1}^{p}\frac{\lambda_i}{(\lambda_i + \alpha + (c - \lambda_p)_+)^2}. \tag{31}$$

For the DP term, Algorithm 1 uses Gaussian noise scale

$$\sigma_{\alpha,c} = \frac{R_x\big(4R_xR_\theta + 2R_y\big)}{n(c+\alpha)} \cdot \frac{\sqrt{2\log(1.25/\delta)}}{\varepsilon},$$

so

$$p\sigma_{\alpha,c}^2 = \frac{2\, p\, R_x^2\big(4R_xR_\theta + 2R_y\big)^2}{n^2(c+\alpha)^2} \cdot \frac{\log(1.25/\delta)}{\varepsilon^2}. \tag{32}$$

If $p/n \to \gamma$, then $p/n^2 = (p/n)^2 \cdot (1/p) = \gamma^2/p$, and thus

$$p\sigma_{\alpha,c}^2 = \frac{2\,\gamma^2\,(R_x^2/p)\,\big(4R_xR_\theta + 2R_y\big)^2}{\varepsilon^2(c+\alpha)^2}\log\Big(\frac{1.25}{\delta}\Big)\cdot(1 + o(1)),$$

which matches the DP-variance term in the first display of Theorem 4.1.

To identify the asymptotic limits of the bias and $\epsilon$-variance terms, we use the assumed Marchenko–Pastur convergence: the empirical spectral distribution of $S$ converges weakly to $\nu_\gamma$ on $[a_\gamma, b_\gamma]$. In addition, the smallest eigenvalue satisfies $\lambda_p \to a_\gamma$, so $(c - \lambda_p)_+ \to (c - a_\gamma)_+$ in the regime covered by the theorem statement. Consequently, replacing $\lambda_p$ by its limit in (30)–(31) and applying the continuous mapping theorem yields

$$\frac{1}{p}\sum_{i=1}^{p}\left(\frac{\alpha + (c - \lambda_p)_+}{\lambda_i + \alpha + (c - \lambda_p)_+}\right)^2 \to \int_{a_\gamma}^{b_\gamma}\left(\frac{\alpha + (c - a_\gamma)_+}{x + \alpha + (c - a_\gamma)_+}\right)^2 \nu_\gamma(dx),$$

and

$$\frac{1}{p}\sum_{i=1}^{p}\frac{\lambda_i}{(\lambda_i + \alpha + (c - \lambda_p)_+)^2} \to \int_{a_\gamma}^{b_\gamma}\frac{x}{(x + \alpha + (c - a_\gamma)_+)^2} \nu_\gamma(dx).$$

Multiplying the latter limit by $p/n \to \gamma$ yields the $\epsilon$-variance term $\int \frac{\gamma x}{(x+\alpha+(c-a_\gamma)_+)^2}\nu_\gamma(dx)$ in the theorem statement.

Finally, under the normalization used in Theorem 4.1, the bias prefactor is written as $R_y^2/p$; plugging this into the limiting expression above and combining with the DP-variance limit from (32) gives the first display of Theorem 4.1.

For the underparameterized setting, the empirical spectrum concentrates near 1 and the MP law collapses to a point mass at 1. Thus the integrals reduce to evaluating the integrands at $x = 1$, giving the stated limits for the bias and $\epsilon$-variance terms, while the DP-variance term follows directly from (32).

We then find the optimal $\alpha$ and $c$. First consider $c \leq 1$. Then $(c - 1)_+ = 0$ and the first two terms depend only on $\alpha$ via $(1 + \alpha)^{-2}$, while the DP term depends on $c$ only through $(c + \alpha)^{-2}$. For any fixed $\alpha$, the DP term is strictly decreasing in $c$, so the best choice over $c \leq 1$ is $c = 1$.

Next consider $c > 1$. In this case $(c - 1)_+ = c - 1$ and we can reparameterize by

$$t := \alpha + (c - 1) \geq c - 1, \qquad 1 + t = c + \alpha.$$

Under this change of variables, all three terms depend on $(\alpha, c)$ only through $t$:

$$g(\alpha, c) = \frac{R_y^2}{p}\left(\frac{t}{1 + t}\right)^2 + \frac{p}{n(1 + t)^2} + \frac{2p\,R_x^2\big(4R_xR_\theta + 2R_y\big)^2}{n^2\varepsilon^2(1 + t)^2}\,\log\!\Big(\frac{1.25}{\delta}\Big).$$

Thus for $c > 1$, minimizing over $\alpha \geq 0$ is equivalent to minimizing the right-hand side over $t \geq c - 1$. Since the unconstrained minimizer over $t \geq 0$ is finite, imposing the additional constraint $t \geq c - 1 > 0$ cannot improve the optimum compared to taking $c = 1$ (which allows all $t \geq 0$). Hence an optimizer must satisfy $c^\star \leq 1$, and therefore $c^\star = 1$.

Now fix $c = 1$. Let

$$A := \frac{R_y^2}{p}, \qquad B := \frac{p}{n} + \frac{2p\,R_x^2\big(4R_xR_\theta + 2R_y\big)^2}{n^2\varepsilon^2}\,\log\!\Big(\frac{1.25}{\delta}\Big).$$

Then

$$g(\alpha, 1) = \frac{A\alpha^2 + B}{(1 + \alpha)^2}.$$

Differentiating for $\alpha \geq 0$ gives

$$\frac{d}{d\alpha}\left(\frac{A\alpha^2 + B}{(1 + \alpha)^2}\right) = \frac{2(A\alpha - B)}{(1 + \alpha)^3},$$

so the unique stationary point is $\alpha^\star = B/A$, which is nonnegative and therefore optimal. Substituting $A, B$ yields the claimed expression for $\alpha^\star$.

For the minimized value, note that at $\alpha^\star = B/A$,

$$A(\alpha^\star)^2 = \frac{B^2}{A}, \qquad (1 + \alpha^\star)^2 = \left(1 + \frac{B}{A}\right)^2 = \frac{(A + B)^2}{A^2}.$$

Hence

$$g(\alpha^\star, 1) = \frac{A(\alpha^\star)^2 + B}{(1 + \alpha^\star)^2} = \frac{\frac{B^2}{A} + B}{\frac{(A+B)^2}{A^2}} = \frac{BA}{A + B}.$$

Expanding $A$ and $B$ gives the stated closed form. $\qquad\square$

### A.5. Proof of Theorem 5.1

*Proof.* Let $Z = \{z_1, \ldots, z_n\}$ and $Z' = \{z_1, \ldots, z_i', \ldots, z_n\}$ be neighboring datasets. Define

$$g(\theta; Z) := \nabla_\theta L_{\alpha,c}(\theta; Z) = \frac{1}{n}\sum_{j=1}^{n}\nabla_\theta f(\theta, z_j) + \alpha_c(Z)\,\theta,$$

where $\alpha_c(Z) = \alpha + \left(c - \frac{1}{n}\sum_{j=1}^{n} h_f(z_j)\right)_+$ and $h_f(z) = \min_{\theta \in \Theta} \lambda_{\min}(\nabla^2 f(\theta, z))$.

Its Jacobian is

$$\nabla_\theta g(\theta; Z) = \frac{1}{n}\sum_{j=1}^{n}\nabla_\theta^2 f(\theta, z_j) + \alpha_c(Z)I \succeq (\alpha + c)\,I,$$

where the last inequality follows from the construction of $\alpha_c(Z)$ (the strong convexity floor is $\alpha + c$).

Let $\hat{\theta} = \hat{\theta}(Z)$ and $\hat{\theta}' = \hat{\theta}(Z')$ be the minimizers of $L_{\alpha,c}(\cdot; Z)$ and $L_{\alpha,c}(\cdot; Z')$, respectively. Then $g(\hat{\theta}; Z) = 0$ and $g(\hat{\theta}'; Z') = 0$. Let $v := \hat{\theta}' - \hat{\theta}$. By the mean value theorem, there exists $\theta_v$ on the line segment between $\hat{\theta}$ and $\hat{\theta}'$ such that

$$g(\hat{\theta}'; Z) - g(\hat{\theta}; Z) = \nabla_\theta g(\theta_v; Z)\,(\hat{\theta}' - \hat{\theta}).$$

Taking inner products with $v$ and using $g(\hat{\theta}; Z) = 0$ yields

$$v^\top g(\hat{\theta}'; Z) = v^\top \nabla_\theta g(\theta_v; Z)\, v \;\geq\; (\alpha + c)\|v\|_2^2.$$

Therefore,

$$(\alpha + c)\|v\|_2^2 \leq \|v\|_2\, \|g(\hat{\theta}'; Z)\|_2 \quad \Rightarrow \quad \|\hat{\theta}' - \hat{\theta}\|_2 \leq \frac{1}{\alpha + c}\,\|g(\hat{\theta}'; Z)\|_2.$$

Since $g(\hat{\theta}'; Z') = 0$, we have

$$\|g(\hat{\theta}'; Z)\|_2 = \|g(\hat{\theta}'; Z) - g(\hat{\theta}'; Z')\|_2.$$

Because $Z$ and $Z'$ differ only at index $i$,

$$g(\hat{\theta}'; Z) - g(\hat{\theta}'; Z') = \frac{1}{n}\Big(\nabla f(\hat{\theta}', z_i) - \nabla f(\hat{\theta}', z_i')\Big) + \big(\alpha_c(Z) - \alpha_c(Z')\big)\hat{\theta}'.$$

By definition of $L_f$,

$$\left\|\nabla f(\hat{\theta}', z_i) - \nabla f(\hat{\theta}', z_i')\right\|_2 \leq L_f.$$

It remains to bound $|\alpha_c(Z) - \alpha_c(Z')|$. First note that the map $t \mapsto (c - t)_+$ is 1-Lipschitz, so

$$|\alpha_c(Z) - \alpha_c(Z')| \leq \left|\frac{1}{n}\sum_{j=1}^n h_f(z_j) - \frac{1}{n}\sum_{j=1}^n h_f(z_j')\right| = \frac{1}{n}\,|h_f(z_i) - h_f(z_i')|.$$

Next, by Weyl's inequality, $\lambda_{\min}(\cdot)$ is 1-Lipschitz w.r.t. operator norm, hence for any $\theta \in \Theta$,

$$\left|\lambda_{\min}(\nabla^2 f(\theta, z_i)) - \lambda_{\min}(\nabla^2 f(\theta, z_i'))\right| \leq \|\nabla^2 f(\theta, z_i) - \nabla^2 f(\theta, z_i')\|_{\mathrm{op}} \leq H_f.$$

Taking $\min_{\theta \in \Theta}$ on both sides gives $|h_f(z_i) - h_f(z_i')| \leq H_f$, and therefore

$$|\alpha_c(Z) - \alpha_c(Z')| \leq \frac{H_f}{n}.$$

Combining the bounds and using $\|\hat{\theta}'\|_2 \leq R_\theta$ (since $\hat{\theta}' \in \Theta$) yields

$$\|g(\hat{\theta}'; Z) - g(\hat{\theta}'; Z')\|_2 \leq \frac{L_f}{n} + \frac{H_f}{n}\,R_\theta = \frac{L_f + H_f R_\theta}{n}.$$

Hence the $\ell_2$ sensitivity of the optimizer satisfies

$$\|\hat{\theta}(Z) - \hat{\theta}(Z')\|_2 \leq \frac{L_f + H_f R_\theta}{n(\alpha + c)}.$$

Finally, Euclidean projection is non-expansive, so

$$\|\Pi_{R_\theta}\hat{\theta}(Z) - \Pi_{R_\theta}\hat{\theta}(Z')\|_2 \leq \|\hat{\theta}(Z) - \hat{\theta}(Z')\|_2 \leq \frac{L_f + H_f R_\theta}{n(\alpha + c)}.$$

Applying the Gaussian mechanism to $\Pi_{R_\theta}\hat{\theta}(Z)$ with noise standard deviation

$$\sigma = \frac{4\sqrt{2\log(1.25/\delta)}\,(L_f + H_f R_\theta)}{\varepsilon n(\alpha + c)}$$

gives $(\varepsilon, \delta)$-DP. $\qquad\square$

## A.6. Implementation details and results for Real Data Analysis

Our method was compared against other methods on 4 real datasets, as outlined in Section 7. The parameter inputs for all algorithms are presented in table 2 and the estimation error $\|\hat{\theta} - \theta_0\|_\Sigma$ over 100 iterations is reported in table 3.

For all methods, covariance proxy $\hat{\Sigma}$ is estimated from the auxiliary dataset, which is 30% of the corresponding full dataset, denoted as $(X_{subset}, y_{subset})$. $\theta_{subset}$ denotes the non-private ridge estimate on the auxiliary dataset.

The other 70% of the data is used for regression.

| Method | Parameter | Estimate |
|---|---|---|
| DADP | $R_x, R_y, R_\theta$ | $\max_{\text{subset}} \|\hat{\Sigma}^{-1/2} X_i\|_2, \max_{\text{subset}} |y_i|, \|\hat{\Sigma}^{1/2} \theta_{\text{subset}}\|_2$ |
| | $\Sigma$ | $\hat{\Sigma} = \text{Cov}(X_{\text{subset}})$ |
| CWZ-HD | $b_x, b_\theta$ | $\max_{\text{subset}} \|X_i\|_2, \|\theta_{\text{subset}}\|_2$ |
| | $L$ | $\max\left(p\lambda_{\max}(\hat{\Sigma}), \frac{1}{p\lambda_{\min}(\hat{\Sigma})}\right)$ |
| | $\sigma$ | subset estimate $\hat{\sigma}_{OLS}$ |
| CWZ-LD | $b_x, b_\theta$ | $\max_{\text{subset}} \|X_i\|_2, \|\theta_{\text{subset}}\|_2$ |
| | $L$ | $\max\left(p\lambda_{\max}(\hat{\Sigma}), \frac{1}{p\lambda_{\min}(\hat{\Sigma})}\right)$ |
| | $\sigma$ | subset estimate $\hat{\sigma}_{OLS}$ |
| JL | covariance of $(X \ Y)$ | $\text{Cov}(X_{\text{subset}}, Y_{\text{subset}})$ |
| | upper bound on $\|X \ y\|_2$ | $\sqrt{\max_{\text{subset}} \|X_i\|_2^2 + \max_{\text{subset}} |y_i|^2}$ |
| Objpert-HD | $\|\theta\|_\infty$ | subset estimate of $b_\theta$ |
| Objpert-LD | $b_x, b_y, b_\theta$ | $\max_{\text{subset}} \|X_i\|_2, \max_{\text{subset}} |y_i|, \|\theta_{\text{subset}}\|_2$ |

*Table 2.* Parameters used for real data analysis

*Table 3.* Error $\|\hat{\theta} - \theta_0\|_\Sigma$ mean (s.d.) for different methods and datasets

| Dataset | $n$ | $p$ | DADP | CWZ-HD | CWZ-LD | JL | Objpert-HD | Objpert-LD |
|---|---|---|---|---|---|---|---|---|
| California | 14448 | 8 | 0.10 (0.03) | 11.79 (16.60) | 10.33 (10.94) | 142934.25 (1014746.11) | 96.41 (4.52) | 0.46 (0.58) |
| energy | 13815 | 27 | 1152.99 (145.34) | 18366.20 (26488.28) | 48230.32 (43200.71) | 5402.12 (7152.41) | 3137.24 (501.16) | 1323.70 (1166.42) |
| wine | 3429 | 11 | 0.17 (0.01) | 1.00 (1.06) | 8.46 (8.35) | 4976.60 (16866.08) | 12.38 (10.08) | 0.82 (0.48) |
| Boston | 355 | 12 | 53.48 (1.29) | 2219.84 (3023.55) | 1933.10 (1878.11) | 57906.71 (138179.47) | 25704.93 (10627.27) | 25508.04 (23382.79) |

DADP achieves the minimum error in all 4 datasets compared to the candidate methods. Observing the large variation in MSE of methods across datasets, we note that the scale of the response variable is different for different datasets. Thus, the MSE values for different datasets are not directly comparable. We summarize the response variable scale, through $var(y)$, in table 4.

*Table 4.* Response variance across datasets

| Dataset | $\text{var}(y)$ |
|---|---|
| California | 0.3239 |
| energy | 10511.353 |
| wine | 0.7844 |
| Boston | 84.5867 |

For datasets **energy** and **Boston**, the response scale is much larger compared to California and wine, which is also reflected in the error magnitudes. To ensure comparability of algorithm performance in different datasets, we present the normalized error $\|\hat{\theta} - \theta_0\|_\Sigma / \text{var}(y)$ in Section 7.

