# OpenReview forum: "Decoupling Regularization and Privacy in Differentially Private Ridge Regression and ERM"
_ICML.cc/2026/Conference — ICML 2026 regular_

### Official Review · Reviewer_5Ubu · 2026-03-07

**Soundness:** 2
**Presentation:** 2
**Significance:** 2
**Originality:** 2
**Overall Recommendation:** 3
**Confidence:** 3

**Summary:**

In this paper the authors develop a method to privately compute the parameters of ridge regression. To this end, the authors focus on the effect of the regularization parameter and its effect when the sample size is small and large. The authors then propose a way to select the parameter given the privacy parameters.

**Compliance With Llm Reviewing Policy:**

Affirmed.

**Final Justification:**

The rebuttal period did not address all my concerns. I acknowledge that my initial review was not entirely accurate, however it seems that "clipping" is carrying a lot of weight in this paper. This is generally fine but it's thus reliant on radii that are not known and need to be either estimated privately. For these reasons I believe this is still borderline.

**Key Questions For Authors:**

I am very unclear on what $\mathcal{G}$ is. What I mean though, is that it is defined in terms of a subscript but a few paragraphs later it switches to a superscript. Even further into the paper both the super and subscript are used. I don't think this is defined anywhere.

In practice and in the simulations, how are the radii selected? How can one do this privately?

I don't understand the clipping of the response variable. This seems to be a flat bound, but this would seem to effect the data near the boundary a lot more than data near the center. That is, data with "large" or "small" x have a much larger probability to have their response clipped if there is truly a linear relationship.

**Limitations:**

Yes

**Strengths And Weaknesses:**

This paper tackles an interesting problem but it leaves much to be desired. While the paper does not need to solve all problems, it seems odd to not even mention LASSO.

Theorem 5.1 is not well stated.

In the decoupling section, its stated that alpha is to guard the worst case, I assume this is meant to say guard AGAINST the worst case?

The "appetizer" example assumes centered data, but this isn't stated. This leaves the question of what else is assumed and not stated.

The proposed solution isn't incredibly significant; the idea seems to boil down to "project your data then proceed as usual" this projection isn't atypical. Standard methods which include standardization and clipping seem to encompass the contribution.

---

> ### Author Rebuttal · Authors · 2026-03-31
>
> Thank you for your comments on this manuscript. We see your concerns about contribution, assumptions, and notations, some of which may come from a misunderstanding. We clarify them below from the most important point (contribution) to the least important notations.
> 1. **Contribution -- "project your data then proceed as usual"**
> We want to clarify that clipping/projection is not the main contribution of our paper. They are standard and useful DP tools to obtain a uniform sensitivity bound. Our main contribution is the *decoupling* of the two roles played by ridge regularization: statistical regularization through $\alpha$, and privacy stabilization through $c$.
> In our method, $c$ is not a standard clipping threshold. It is a dataset-level curvature guardrail implemented through eigenvalue boosting of the Gram matrix. This leads to the risk decomposition in Theorem 3.3 and the tuning analysis in Sections 3-4, which are our core technical contributions. Section 2 is used as an appetizer to illustrate why a single parameter $\alpha$ creates a contradiction in DP ridge regression: typical datasets are well behaved, but DP requires control of the global sensitivity, including pathological datasets with very small $\sum_i x_i^2$. Our two-parameter framework resolves exactly this contradiction. Standard clipping/projection alone does not.
> 2. **Model assumption and theorem statement**
> The reviewer regards "Theorem 5.1 is not well stated" and "the 'appetizer' example assumes centered data, but this isn't stated". We address both points in this reply.
> For a DP estimator, there are two different theoretical questions: *1) whether the procedure is private, 2) whether its prediction loss can be controlled or characterized.*
> For 1), **the privacy guarantee cannot rely on distributional assumptions**, because DP must hold for all possible datasets, including atypical or adversarial ones. **For 2), assumptions are typically needed**, since any analysis of prediction loss requires a probabilistic or statistical model.
>
>     For the **appetizer**,  its purpose is not to establish the later prediction-risk theory, but to isolate the basic contradiction in DP under the simplest possible ridge regression setting. The key point is that, although under typical realizations one expects $\sum_i x_i^2$ to be of order $n$, DP must also account for pathological datasets where $\sum_i x_i^2$ is very small, which makes the sensitivity much larger. This is exactly the contradiction that motivates our decoupling idea through $(\alpha,c)$.
>
>     Therefore, the decoupling argument in the appetizer does not depend on whether the data are centered. Centering is only needed later in Sections 3--4 for the prediction-risk analysis, where the assumptions are stated explicitly. To avoid confusion, we will revise the appetizer to make its role and scope more explicit.
>     For **Theorem 5.1**, the theorem is intended only to establish the DP guarantee for the general ridge-regularized ERM extension. It is not intended to provide an explicit prediction-risk characterization. In that sense, Theorem 5.1 does not need any explicit assumptions. In the revision, we will restate it more explicitly, add a corresponding algorithm, and refer to that algorithm directly..
> 3. **Clipping: radii selection and response clipping**
> The choice of *radii* is a general issue in DP regression methods, not specific to our algorithm. In practice, the radii can be set using prior knowledge or estimated from auxiliary/public data through high empirical quantiles, so that clipping is activated only on rare tail events. This is also the practical approach we use.
> The *response clipping* is naturally derived due to $R_x$, $R_{\theta}$, and the noise. We agree that observations with larger responses are more likely to be affected. Our point is not that clipping is harmless for every sample, but that it is chosen conservatively so that clipping is rare, while still ensuring a valid sensitivity bound and keeping its overall effect on the loss small.
> 4. **LASSO**
> We agree that LASSO is a relevant direction in high-dimensional regression. Our focus here is ridge regression and ridge-regularized ERM, where the closed-form structure allows an explicit sensitivity analysis and the bias-variance-DP-variance decomposition. LASSO further involves feature selection, which brings additional DP questions beyond the scope of this paper. We will add discussion to better position our contribution relative to LASSO and sparse-regression settings
> 5. **Notation and wording**
> For $\mathcal{G}$, we define it with a subscript to denote the parameter $c$ chosen. The corresponding complement is denoted as $\mathcal{G}^c$, which is a standard use. We understand it may cause confusion and will revise it.
>     We will also correct the phrase “guard the worst case” to “guard against the worst case.” Here, we want to clarify that it is $c$, not $\alpha$, that guards against the worst case.

---

> > ### Author Rebuttal · Reviewer_5Ubu · 2026-04-01
> >
> > Thank you for your responses.
> >
> > I'll take the responses in consideration for score adjustment.
> >
> >
> > Thanks for the latter comments. I just reopened the paper and I don't see $\mathcal{G}^c$ ever being defined. Given the explanation I am still confused. Is the $c$ playing the role of the complement or do the authors mean that by making $c$ a superscript that implies the complement? So is  $\mathcal{G}^c$ meant to be  $\mathcal{G}_c^c$ where the lower refers to the set and the upper is the standard complement notation?

---

> > > ### Author Response · Authors · 2026-04-01
> > >
> > > Thank you for your recognition of our replies. We deeply appreciate your timely reply!
> > >
> > > We apologize for the confusing notations about $\mathcal{G}$. We use $c$ in the subscript as a parameter in the good set definition, and $c$ in the superscript as "complement". We realize that it may be confusing.
> > >
> > > We have double-checked its appearances in the paper. Taking the univariate case as an example, we list all the possible forms here and our promised revisions:
> > > 1. $\mathcal{G}_c$: already defined as $\sum x_i^2 \geq c n$, no confusion.
> > > 2. $\mathcal{G}_c^{c}$: the complement of $\mathcal{G}_c$.
> > > When it appears, we give the equivalent saying that $\sum x_i^2 < c n$. In the revised version, we will remove all such notations. Instead, we will use the phrase "$Z \notin \mathcal{G}_c$".
> > > 3. $\mathcal{G}^{c}$: we missed the subscript $c$ here as a typo. It should be the same with $\mathcal{G}_c^{c}$ in part 2.
> > >  In the revised version, we will use "$Z \notin \mathcal{G}_c$" and remove such notations.
> > >
> > > For the multivariate case, we will do the same checking and replacements.
> > >
> > > Thank you again for your careful reading. We believe that will improve our presentation.

---

### Official Review · Reviewer_mSmk · 2026-03-09

**Soundness:** 3
**Presentation:** 4
**Significance:** 3
**Originality:** 3
**Overall Recommendation:** 5
**Confidence:** 4

**Summary:**

In differentially private ridge regression with panelty parameter $\alpha$, there exists a contradiction: statistically a smaller $\alpha$ is prefered but from the perspective of worst-case DP, a larger $\alpha$ is needed to control sensitivity. By analysis of univariate ridge regression, the authors found the important role that $\sum_j x_j^2$ played and proposed to add $cn$ to the denominator to stabilize the estimator when $\sum_j x_j^2$ is unacceptably small. In this way they proposed the decoupled adaptive DP ridge estimator. This paper is both well-motivated and well-written. The analysis of the univariate case at the beginning immediately delivers the core idea to the readers.

**Compliance With Llm Reviewing Policy:**

Affirmed.

**Key Questions For Authors:**

1.	In Section 4, the discussion about the choice of $\alpha$ and $c$ still relies on specific model assumptions such as random-design model and Guassian prior on the signal. Is it possible to more generally discuss the fine tuning of these parameter? Or what are the difficulties in doing so?
2.	In Section 3.2, the authors assume the covariance proxy $\Sigma$ is estimated from external dataset (public or auxiliary datasets).  Then are the authors actually requiring $\Sigma$ be independent of samples $x_i$? Why not estimate $\Sigma$ as the empirical covariance of $x_i$?
3.	Still in Section 3.2, the authors assume $x_i\sim N(0,\Sigma_x)$. Why restrict the distribution of $X$ to Guassian? Can the results in Section 3.2 apply to more general settings?

**Limitations:**

The proposed framework relies on several assumptions (e.g., known or proxy covariance, Guassian distribution and bounded data) that may not hold in more realistic settings, which could affect its robustness in practice.

**Strengths And Weaknesses:**

•	Soundness:  The decoupling strategy used in this paper is intuitive and effective. The decoupled adaptive DP ridge estimator guarantees privacy protection, which is theoretically proved.
•	Presentation: The paper is clearly written and well structured.
•	Significance: The method introduced in this paper addresses the contradiction about the penalty parameter $\alpha$ in ridge regression. And the method is easy to adopt by others in practice.
•	Originality: This paper introduced an innovative trick that can be effectively adopted in the field of DP ridge regression, which is of much practical value.

---

> ### Author Rebuttal · Authors · 2026-03-31
>
> Thank you for your careful reading and for the thoughtful questions. The three questions are all on the generalization problem. A quick answer is: the DP guarantee itself does not rely on Gaussianity, and the Gaussian/random-design assumptions are mainly used for the utility analysis. The key is to control the lower tail of $\lambda_{\min}(\tilde X \tilde X^\top/n)$ and hence the choice of $c$. Given the current random matrix results, it can be easily extended to sub-Gaussian distributions, but more generalized settings need careful discussion. We discuss each comment in detail below. We also added real-data experiments, which show the superior performance of our method beyond the synthetic Gaussian setting.
>
> 1. **On the choice of $\alpha$ and $c$ in Section 4**
>    Section 4 aims at an exact asymptotic risk formula, not only an upper bound. For this, we need a tractable limit for the empirical spectrum of $\tilde X \tilde X^\top/n$ and a precise understanding of the lower spectral edge, to analyze the boosting term depending on ${(c-\lambda_{\min})}_+$. In the current paper, this is done through Marchenko--Pastur theory, so Gaussian design, or at least sub-Gaussian concentration, is needed.
>
>     In a more general setting beyond sub-Gaussian distributions, the decoupling idea still applies. But how to determine $c$, so that the bad event ${\lambda_{\min}<c}$ has a small probability and the good event enjoys a stable solution, needs case-by-case analysis. Our suggestion is to fix the range of the tuning parameters $c$ and $\alpha$, instead of deriving an explicit formula. It might be at the cost of optimality.
>
> 2. **On why $\Sigma$ is taken from external / auxiliary data**
>    Our analysis is carried out after whitening, with $u_i=\Sigma^{-1/2}x_i$, so the distribution of the whitened design is central in the proofs. When $\Sigma$ is fixed externally, the resulting matrix has the random-matrix structure (independent entries) used in the risk decomposition and in controlling $\lambda_{\min}$. If $\Sigma$ is estimated from the same sample, the whitened design have *dependent* entries, then the random matrix theory breaks, and the utility analysis no longer works. This is mainly a utility-analysis issue, not a fixed-$\Sigma$ privacy issue.
> However, since this dependence is weak, usually it doesn't hurt much in practice, while in theory the dependence is tricky. The only concern may be the privacy in $\Sigma$. We may use sub-samples or $\hat\Sigma$ with additional noise to protect DP.
>
> 3. **On the Gaussian design assumption in Section 3.2**
>    The Gaussian design assumption is mainly used to justify the choice of $c$: we need ${\lambda_{\min}(\tilde X \tilde X^\top/n)<c}$ to be rare, while on the complementary event the sensitivity is controlled by $c+\alpha$. In the current paper, this is done through random matrix theory for Gaussian entries. With this in mind, we expect the analysis to extend at least to sub-Gaussian designs, since analogous lower-edge concentration results are available there.
> More generally, if one has good control of the lower tail of $\lambda_{\min}$ for a broader design class, the same decoupling principle should still apply.
>
> To complement the theory, we also added real-data experiments on four regression datasets: Communities and Crime [1], Wine Quality [2], Appliances Energy Prediction [3], and Boston Housing [4]. Under the same auxiliary-data protocol used for all methods, DADP achieves the best mean MSE on all four datasets.
>
> **Table 1. Mean MSE (s.d.) over 100 runs with $\epsilon = 0.5$ and $\delta = 10/n$.**
>
> | Dataset     |   $n$ | $p$ |   DADP (Ours) |          CWZ-HD |          CWZ-LD |              JL |      Objpert-HD |      Objpert-LD |
> | ----------- | ----: | --: | ------------: | --------------: | --------------: | --------------: | --------------: | --------------: |
> | crime [1]   |  1396 | 100 |   0.02 (0.00) |     0.10 (0.16) |     0.12 (0.14) | 680.75 (2.48e3) |     6.78 (4.11) |     0.06 (0.07) |
> | wine [2]    |  3429 |  11 |   0.09 (0.00) |    6.98 (11.26) |   22.02 (24.82) | 2.85e4 (1.96e5) |   16.34 (12.31) | 231.26 (647.00) |
> | energy [3]  | 13815 |  27 | 1.38e3 (0.29) | 2.41e3 (1.24e3) | 9.36e4 (7.04e4) | 1.77e4 (1.18e5) | 4.45e3 (825.71) | 1.06e5 (3.71e4) |
> | housing [4] |   355 |  12 |  41.37 (0.00) | 2.75e3 (3.01e3) | 2.77e3 (2.85e3) | 8.72e4 (2.30e5) | 2.77e4 (1.02e4) | 1.69e8 (3.03e8) |
>
> [1] Michael Redmond. *Communities and Crime*. UCI Machine Learning Repository, 2002.
>
> [2] Paulo Cortez, A. Cerdeira, F. Almeida, T. Matos, and J. Reis. *Wine Quality*. UCI Machine Learning Repository, 2009.
>
> [3] Luis Candanedo. *Appliances Energy Prediction*. UCI Machine Learning Repository, 2017.
>
> [4] Harrison, D. and Rubinfeld, D.L. *Boston housing dataset*. CMU StatLib, 1978. Accessed: 2026-03-29.

---

> > ### Author Rebuttal · Reviewer_mSmk · 2026-04-03
> >
> > My concers have been mainly addressed.

---

### Official Review · Reviewer_sDBs · 2026-03-11

**Soundness:** 3
**Presentation:** 3
**Significance:** 3
**Originality:** 3
**Overall Recommendation:** 4
**Confidence:** 3

**Summary:**

Differentially private (DP) ridge regression faces an inherent tension: a large regularization parameter $\alpha$ stabilizes the optimization and reduces sensitivity, thereby lowering the required noise scale, but also introduces estimation bias and degrades predictive accuracy. This paper addresses this tension by proposing a decoupling of $\alpha$ into two distinct parameters: (i) a penalty parameter $\alpha$ that governs the regularized solution, and (ii) a worst-case guard parameter $c$ that enforces a sensitivity bound to control the DP noise scale. The authors conduct a rigorous theoretical analysis of the expected prediction loss under three representative regimes. Optimal parameter choices are derived under explicit assumptions on the training data and parameters. The decoupling framework is further extended to ridge-regularized empirical risk minimization (ERM). Simulation experiments demonstrate that the proposed algorithm achieves lower prediction loss compared to several baselines.

**Compliance With Llm Reviewing Policy:**

Affirmed.

**Final Justification:**

Based on the discussion, I decide to maintain my positive score.

**Key Questions For Authors:**

How should practitioners select parameters in real-world settings? It is unclear how a practitioner should select these parameters in real-world deployment, and whether data-driven or adaptive selection strategies would preserve the theoretical guarantees.

**Limitations:**

Yes

**Strengths And Weaknesses:**

Strengths:
- The paper presents a clear and well-motivated intuition for the proposed decoupling strategy.
- The theoretical analysis is thorough and technically sound, covering different representative regimes with characterizations of the prediction loss and optimal parameter choices.
- Simulation results are consistent with the theoretical predictions and support the practical advantage of the proposed method over existing baselines.

Weaknesses:
- The empirical evaluation is limited to synthetic simulations. The absence of experiments on real-world datasets makes it difficult to assess the practical utility and robustness of the proposed method under realistic data distributions.
- It is unclear whether the distributional assumptions underlying the theoretical analysis about optimal parameter choice are likely to hold for real-world datasets. There is no guidance on data-driven or adaptive strategies for parameter selection in practice, where the true data distribution or parameter prior is unknown. A discussion of how practitioners should choose parameters in applied settings would strengthen the paper.
- The extension to ridge-regularized ERM is supported only by theoretical analysis, without any accompanying empirical evaluation.

---

> ### Author Rebuttal · Authors · 2026-03-30
>
> Thank you for these helpful comments on the practical side. Below, we add several real-world datasets, clarify our parameter choices and the assumptions, and explain more clearly the scope of the ERM extension.
>
> 1. **Real-world Experiments.**
> We compared our method with other methods on 4 real regression datasets from the UCI Machine Learning Repository and the CMU StatLib repository: Communities and Crime [1], Wine Quality [2], Appliances Energy Prediction [3], and Boston Housing [4]. For each dataset, we take 30% as auxiliary data (not private) and the other 70% as the dataset for regression. For all methods, the covariance proxy and boundary parameters are estimated from this auxiliary data. We report the non-private ridge estimator as the baseline for error computation.
>
> **Table 1. Mean MSE (s.d.) over 100 runs with $\epsilon = 0.5$ and $\delta = 10/n$.**
>
> | Dataset | $n$ | $p$ | DADP (Ours) | CWZ-HD | CWZ-LD | JL | Objpert-HD | Objpert-LD |
> |---|---:|---:|---:|---:|---:|---:|---:|---:|
> | crime [1] | 1396 | 100 | 0.02 (0.00) | 0.10 (0.16) | 0.12 (0.14) | 680.75 (2.48e3) | 6.78 (4.11) | 0.06 (0.07) |
> | wine [2] | 3429 | 11 | 0.09 (0.00) | 6.98 (11.26) | 22.02 (24.82) | 2.85e4 (1.96e5) | 16.34 (12.31) | 231.26 (647.00) |
> | energy [3] | 13815 | 27 | 1.38e3 (0.29) | 2.41e3 (1.24e3) | 9.36e4 (7.04e4) | 1.77e4 (1.18e5) | 4.45e3 (825.71) | 1.06e5 (3.71e4) |
> | housing [4] | 355 | 12 | 41.37 (0.00) | 2.75e3 (3.01e3) | 2.77e3 (2.85e3) | 8.72e4 (2.30e5) | 2.77e4 (1.02e4) | 1.69e8 (3.03e8) |
>
> [1] Michael Redmond. *Communities and Crime*. UCI Machine Learning Repository, 2002. DOI: 10.24432/C53W3X.
> [2] Paulo Cortez, A. Cerdeira, F. Almeida, T. Matos, and J. Reis. *Wine Quality*. UCI Machine Learning Repository, 2009. DOI: 10.24432/C56S3T.
> [3] Luis Candanedo. *Appliances Energy Prediction*. UCI Machine Learning Repository, 2017. DOI: 10.24432/C5VC8G.
> [4] Harrison, D. and Rubinfeld, D.L. *Boston housing dataset*. CMU StatLib, 1978. Accessed: 2026-03-29.
>
> 2. **Parameter selection in real datasets.**
> To implement our algorithm, we need the following parameters: estimated $\Sigma$, $\alpha$, $c$, and Radii ($R_X$, $R_y$).
> - Radii. The choice of radii is a general issue for DP regression methods (e.g. CWZ, JL), not unique to our algorithm. In practice, $R_X$ and $R_y$ can be set using prior knowledge or estimated from a public/auxiliary dataset through high quantiles. This is also the approach we adapted in real data analysis.
> - $\Sigma$. In our algorithm, it should be achieved using auxiliary data or prior knowledge. This is exactly the setting assumed in Section 3.2.
> - $\alpha$ and $c$. The choice of $\alpha$ and $c$ largely depends on the random matrix theory, so we put them together to explain.
> The choice of $c$ is based on the random matrix theory that characterizes the smallest eigenvalue of $\tilde{X} \tilde{X}^\top$.
> Currently, the choice of $c$ is motivated by random matrix theory, which characterizes the performance of the smallest eigenvalue. This theory holds when $\tilde{X}$ has sub-Gaussian entries. Hence, we only need to check the tail in $\tilde{X}$ and we can apply the $c = (1 - \sqrt{\gamma})^2$.
> The choice of $\alpha$ depends on $c$ and random matrix theory. a). When $n \gg p$, then we have an optimal $\alpha$ in Theorem 4.1. Same as the choice of $c$, it can be applied when $\tilde{X}$ has sub-Gaussian tails. b). When $p/n \to \gamma < 1$, Theorem 4.1 expresses the risk in a complicated way. It is so complicated that we do not have an explicit form of optimal $\alpha$. Hence, instead of the "otpimal $\alpha$", we suggest to use a good-enough $\alpha$ in Theorem 3.5. The results hold for sub-Gaussian entries in $\tilde{X}$.
>
> In our empirical analysis above, we chose $\alpha$ and $c$ according to the formula in our paper. We did not use an adaptive strategy.
>
> 3. **On the ERM extension.**
>    In the current submission, we only have the theoretical results for the ERM extension. We hope to introduce this decoupling method and focus on the ridge regression as a showcase. The ERM section is intended as a conceptual and theoretical generalization of the same decoupling principle. We will revise the text to make this scope clearer.

---

> > ### Author Rebuttal · Reviewer_sDBs · 2026-04-04
> >
> > Thank you for your response. Could you clarify why some real-world experiments exhibit notably large mean MSE and standard deviation?

---

> > > ### Author Response · Authors · 2026-04-04
> > >
> > > Thank you for your response and the follow-up questions. We are very happy to have the chance to explain this phenomenon, which was regrettably omitted in our last response due to the 5000-character limit.
> > >
> > > &nbsp;
> > >
> > > ### Large MSE due to Large Response Scale ####
> > > For the calculation of MSE and perturbed estimators, the scale of the response variable $y$ plays an important role. When the scale $R_y$ varies substantially across datasets, the MSE values are not directly comparable.  Table 1 summarizes $R_y$ and $var(y)$ across datasets, where the energy dataset has a very large $R_y$ and $var(y)$.
> > >
> > > **Table 1. Summary of response scales**
> > >
> > > | Dataset | $\mathrm{var}(y)$ | $\mathrm{sd}(y)$ | $R_y$ |
> > > |---|---:|---:|---:|
> > > | crime | 5.428e-2 | 0.2330 | 0.7620 |
> > > | wine | 7.844e-1 | 0.8856 | 3.1221 |
> > > | **energy** | **1.051e4** | **102.52** | **752.31** |
> > > | housing | 8.459e1 | 9.1971 | 27.4672 |
> > >
> > > From Table 2, the response scale on **energy** is much larger than on the other datasets. This inflates the absolute MSE of all methods on that dataset, so the raw values should not be interpreted in isolation.
> > >
> > > To separate this dataset-scale effect from the algorithmic behavior, we additionally normalize the real-data MSE by $\mathrm{var}(y)$. The resulting values are shown in Table 2.
> > >
> > > **Table 2. Mean MSE/$\mathrm{var}(y)$ (s.d./$\mathrm{var}(y)$) over 100 runs**
> > >
> > > | Dataset | n | p | DADP (Ours) | CWZ-HD | CWZ-LD | JL | Objpert-HD | Objpert-LD |
> > > |---|---:|---:|---:|---:|---:|---:|---:|---:|
> > > | crime | 1396 | 100 | 0.368 (0.000) | 1.842 (2.948) | 2.211 (2.579) | 1.254e4 (4.569e4) | 124.9 (75.72) | 1.105 (1.290) |
> > > | wine | 3429 | 11 | 0.115 (0.000) | 8.899 (14.356) | 28.074 (31.644) | 3.634e4 (2.499e5) | 20.83 (15.69) | 294.84 (824.88) |
> > > | **energy** | **13815** | **27** | **0.1313 (2.76e-5)** | **0.2293 (0.1180)** | **8.905 (6.698)** | **1.684 (11.226)** | **0.4234 (0.0786)** | **10.08 (3.530)** |
> > > | housing | 355 | 12 | 0.4891 (0.000) | 32.51 (35.58) | 32.75 (33.69) | 1030.9 (2719.1) | 327.47 (120.59) | 1.998e6 (3.582e6) |
> > >
> > > After this normalization, the unusually large absolute error on **energy** is no longer anomalous: our method has normalized error 0.1313 on **energy**, compared with 0.115 on wine, 0.368 on crime, and 0.489 on housing. This indicates that the large absolute MSE on **energy** is primarily a response-scale effect rather than a sign of unusually poor covariate conditioning.
> > >
> > > We didn't present this table in the first response, because this table cannot be directly interpreted as "error". It only suffices to understand the error magnitude.
> > >
> > > &nbsp;
> > >
> > > ### Abnormal Covariates cause Large MSE ####
> > > For **housing**, the relevant phenomenon is different. Even after normalization, our method remains controlled, whereas several baselines are still extremely large. This is consistent with the geometry of the dataset: 1. heterogeneous feature scales; 2. a severely ill-conditioned auxiliary covariance proxy; and 3. $n/p$ is small. In this regime, methods without explicit covariance-aware stabilization are much more sensitive to the poor geometry, while our method remains more stable because it whitens the covariates using the covariance proxy and enforces a curvature floor through the decoupled stabilization parameter $c$.
> > >
> > > &nbsp;
> > >
> > > Therefore, the notably large mean MSE and sd are due to two reasons: **energy** mainly reflects a **response-scale effect**, while **housing** mainly reflects **poor covariate geometry**. Our method remains stable in both regimes, and the advantage is especially clear on housing, where competing methods are much less robust.
> > >
> > > In the revised paper, we will consider some other possible measurements to avoid this confusion.

---

### Official Review · Reviewer_3shM · 2026-03-16

**Soundness:** 3
**Presentation:** 3
**Significance:** 2
**Originality:** 3
**Overall Recommendation:** 4
**Confidence:** 3

**Summary:**

The paper studies ridge regression and ridge-regularized empirical risk minimization under $(\varepsilon,\delta)$-differential privacy via output perturbation. Its main idea is to decouple the usual ridge tuning parameter into a statistical penalty $\alpha$, which defines the target ridge/ERM solution, and a privacy-stabilization parameter $c$, which enforces a curvature floor / boosts the minimum eigenvalue and yields a tighter sensitivity bound. The paper then derives a bias–variance–DP-variance decomposition, gives tuning rules for several $(n,p)$ regimes, and claims improved accuracy over single-parameter regularization, with an extension of the same decoupling principle to general ridge-regularized ERM.

**Compliance With Llm Reviewing Policy:**

Affirmed.

**Final Justification:**

I will keep my positive score.

**Key Questions For Authors:**

See Weakness

**Limitations:**

Yes

**Strengths And Weaknesses:**

## Strengths
1. **The paper isolates a real and important tension in private ridge regression.**
   The introduction clearly explains that, under DP, the usual ridge parameter simultaneously controls statistical bias/variance and global sensitivity, which makes tuning a privacy–accuracy bottleneck. The proposed split into $\alpha$ and $c$ is conceptually simple and easy to understand from the univariate appetizer through the multivariate construction.
2. The paper gives an explicit boosting operator $Q \vee cI$, a stabilized estimator, a sensitivity bound in Lemma 3.2, and a full Algorithm 1 with clipping, projection, and Gaussian perturbation. This makes the proposal implementable and technically transparent.
3. Theorem 3.3 gives a useful decomposition that makes the role of $c$ explicit. The risk is decomposed into bias from ridge + boosting, sampling variance, and DP variance.

## Weaknesses
1. **There is a real proof problem in Theorem 3.5 as written.**
   In the proof, the authors define the no-clipping event $E$, explicitly proceed “on $E$,” and then state that $\tilde X \tilde X^\top \sim \text{Wishart}(A,n)$ on $E$. That is not correct: conditioning on $E$ changes the law, precisely because $E$ is defined through maxima of $\|u_i\|_2$ and $|y_i|$. Therefore the subsequent identity $E[(\tilde X\tilde X^\top)^{-1}] = A^{-1}/(n-p-1)$ does not follow from the argument given.
2. **The proportional-asymptotics analysis appears internally inconsistent for $\gamma \ge 1$.**
   Theorem 4.1 defines the effective regularization as $\lambda(\alpha,c)=\alpha+(c-a_{\gamma}) $
 and claims an optimal $c=a_\gamma=(1-\sqrt{\gamma})^2$ for $p/n\to\gamma\in(0,\infty)$. But in the proof the authors themselves note that the smallest eigenvalue satisfies $\lambda_p\to 0$ for $\gamma\ge 1$, and then replace $(c-\lambda_p)$ by $(c-a_{\gamma})$. For $\gamma > 1$, $a_{\gamma}>0  $ while $\lambda_p\to 0 $ because of the null-space mass; these are not interchangeable. This undermines the claimed “exact characterization” and the optimal-$c$ claim in the overparameterized regime unless the theorem is explicitly restricted to $\gamma<1$.

---

> ### Author Rebuttal · Authors · 2026-03-30
>
> Thank you for your careful reading of this manuscript and for pointing out these two mistakes. We deeply appreciate it.
>
> Before the detailed reply below, we want to have a quick reply on the two problems: the proof for Theorem 3.5 on the "conditional Wishart dist." problem can be solved with some techniques, and the proof for $\gamma \geq 1$ is our mistake. It needs new formula.
>
> 1. For Theorem 3.5, the current conditional-Wishart step is not proper. The most common way to deal with this problem is as follows:
> - a. Decompose the outcome space into $E$ and $E^c$, where $P(E^c) \lesssim 1/n$.
> - b. On the event $E$, we have that $E[(\tilde{X}\tilde{X}^{\top})^{-1}*1_E] \leq A^{-1}/(n - p - 1)$. Here we are not using the conditional distribution.
> - c. On the event $E^c$, instead of checking the trace, we check the loss directly. Note that both $\tilde\beta$ and $\beta^* $ are bounded, so the loss is also bounded. Therefore, $E[loss*1_{E^c}] \leq const * Pr(E^c) \lesssim 1/n$.
>
>     Combining these and we have the result. In the camera-ready version, we will revise the proof.
>
> 2. For Theorem 4.1, we thank you very much for pointing out this problem. Our initial proof was only for the case that $\gamma < 1$. Later, we developed the result for $\gamma > 0$ in Theorem 3.5 (we have carefully checked), and used the same condition for Theorem 4.1 without careful re-checking.
>
>     When $\gamma > 1$, the null-space mass implies $\lambda_{\min}\to 0$, so the effective regularization becomes $\alpha + c$ (different from the case that $\gamma < 1$). The risk decomposition formula in Theorem 4.1 still holds, with $\lambda(\alpha, c) = \alpha + c$. However, now $\alpha$ and $c$ are entangled, and we do not have the optimal choice of $c$. In the camera-ready version, we will revise the theorem statement and proofs accordingly.
>
>     We want to point out that this entanglement is natural in the overparameterized setting. In the overparameterized setting, the rank of $XX^\top$ is $n < p$, and the ridge parameter $\alpha$ must be large to reduce the multicollinearity.

---

> > ### Author Rebuttal · Reviewer_3shM · 2026-04-04
> >
> > Thanks for your reply, I will keep my score.

---

### Decision · Program_Chairs · 2026-04-30

**Decision:**

Accept (regular)

**Comment:**

The paper considers ridge regression and ridge-regularized empirical risk minimization under approximate differential privacy, focusing on output perturbation mechanisms. The authors aim to make the role of regularization explicit in the DP setting by decoupling it into two parameters: a statistical penalty parameter that defines the target ridge solution, and a worst-case stabilization parameter that controls the global sensitivity and, in turn, the scale of the DP noise added to the output. The paper supports this formulation with both theoretical guarantees and numerical experiments.

The reviewers were tepid but favorable about the submission. Reviewer 3shM noted that the paper isolates a real and important tension in private ridge regression, but identified issues in the proofs of Theorems 3.5 and 4.1 that did not match the statements of the main theorems. The authors acknowledged these issues in their rebuttal, corrected the proof in their rebuttal, and committed to revise the statement in a final version. After discussion, the reviewer acknowledged their concerns as fully resolved.

Reviewer sDBs also had an initial positive score though noted that the empirical evaluation was limited to synthetic exiperiments. The authors responded with experiments on several real-world regression datasets comparing against several benchmarks, which led the reviewer to maintain their positive score. Reviewer mSmk was the most positive of the four, appreciating the decoupling strategy introduced by the paper and finding the univariate appetizer particularly effective at conveying the core idea.

Reviewer 5Ubu maintained a negative score after the rebuttal. They acknowledged that parts of their initial review were not entirely accurate, but remained concerned about the role of clipping and the reliance on quantities that would need to be estimated privately. I note that I read the authors' AC-confidential comment regarding this discussion.

Overall, I found the paper to be well-written and a solid contribution to an important problem in DP. Despite the significant progress made in DP over the past years, even simple regression problems, like ridge regression, suffer  utility loss under naive application of approximate DP mechanisms. This paper takes an important step toward helping us understand how to design better DP regression.